# Global projections of future urban land expansion under shared socioeconomic pathways

Guangzhao Chen [1,9], Xia Li [2,9]*, Xiaoping Liu[1,3]*, Yimin Chen[1]*, Xun Liang[4], Jiye Leng[1,5], Xiaocong Xu[1], Weilin Liao[1], Yue'an Qiu [1,6], Qianlian Wu[1,7] & Kangning Huang [8]

Despite its small land coverage, urban land and its expansion have exhibited profound impacts on global environments. Here, we present the scenario projections of global urban land expansion under the framework of the shared socioeconomic pathways (SSPs). Our projections feature a fine spatial resolution of 1 km to preserve spatial details. The projections reveal that although global urban land continues to expand rapidly before the 2040s, China and many other Asian countries are expected to encounter substantial pressure from urban population decline after the 2050s. Approximately 50–63% of the newly expanded urban land is expected to occur on current croplands. Global crop production will decline by approximately 1–4%, corresponding to the annual food needs for a certain crop of 122–1389 million people. These findings stress the importance of governing urban land development as a key measure to mitigate its negative impacts on food production.

---

[1] Guangdong Provincial Key Laboratory of Urbanization and Geo-simulation, School of Geography and Planning, Sun Yat-sen University, 135 West Xingang Road, Guangzhou 510275, China. [2] Key Lab of Geographic Information Science (Ministry of Education), School of Geographic Sciences, East China Normal University, 500 Dongchuan Road, Shanghai 200241, China. [3] Southern Marine Science and Engineering Guangdong Laboratory (Zhuhai), 9 Jintang Road, Xiangzhou, Zhuhai 519000, China. [4] School of Geography and Information Engineering, China University of Geosciences, 68 Jincheng Rd., Wuhan, Hubei 430078, China. [5] Department of Geography and Planning, University of Toronto, Toronto, ON M5S3G3, Canada. [6] Faculty of Geographical Science, Beijing Normal University, No.19 Xinjiekou Outer St, Beijing 100875, China. [7] School of Geography and Ocean Science, NanJing University, 163 Xianlin Avenue, Nanjing 210023, China. [8] Yale School of Forestry and Environmental Studies, 380 Edwards Street, New Haven, CT 06511, USA. [9] These authors contributed equally: Guangzhao Chen, Xia Li. *email: lixia@geo.ecnu.edu.cn; liuxp3@mail.sysu.edu.cn; chenym49@mail.sysu.edu.cn

Urban land covers only a small proportion of the global terrestrial surface but is home to more than half of the world's population[1]. Urban land is expanding even faster than urban population[2], exerting a profound impact on biodiversity conservation and the water, carbon, aerosol and nitrogen cycles in the climate system at the local and global scales[3–5]. Urban areas contribute to 70% of global anthropogenic greenhouse gas emissions[6]. Urban expansion has resulted in more than 80% of natural habitat loss in local areas[7]. Therefore, a proper understanding of how future urban land change will affect other land covers is important to alleviate the social and environmental problems that challenge the sustainable developments of human societies. Here, we present spatially explicit projections of global urban land expansion from 2015 to 2100 and discuss its impacts. The open-access dataset of these projections is available via https://doi.pangaea.de/10.1594/PANGAEA.905890 OR http://www.geosimulation.cn/GlobalSSPsUrbanProduct.html.

Projections of urban land patterns require established scenarios that represent possible future socio-economic and environmental conditions. For instance, a 2030 global urban land map has been developed based on global population and economy predictions made by the United Nations[8]. The universal climate scenarios developed by the Intergovernmental Panel on Climate Change (IPCC) have also been used to simulate future changes in global and regional land covers, including an urban land category[9–11]. However, some models only provide a single scenario based on the historical trajectory in the simulation of future global urban growth[12]. The simulation based on a single scenario could hinder the applications in climate and environmental change studies. In our study, we carry out scenario simulations based on the shared socioeconomic pathways (SSPs), mainly because SSPs provide a more comprehensive framework by considering the potential pathways and uncertainties of future socio-economic factors. The SSPs describe how global society, demographics and economics will change in the coming century concerning policy assumptions and the socio-economic narrative (e.g., energy demand and supply, and technological change)[13,14]. The SSPs allow us to conduct unified and comparable multi-scenario urban simulations[13,15,16].

The SSP framework has five scenarios. SSP1 is a sustainable pathway that is people oriented[17] and uses green roads. SSP2 is a middle pathway between SSP1 and SSP3[18]. SSP3 is a regional rivalry pathway contrary to global cooperation[19]. SSP4 is a divided pathway in which inequality and stratification are increasing both across and within countries[20]. SSP5 is a fossil-fuelled development pathway in which the global economy grows rapidly, but people face severe mitigation challenges[21].

The SSP framework is a critical component in the ongoing IPCC assessment of global climate change[16,22]. Although some recent studies have predicted urban land distribution under SSPs at regional scales[23], relevant results at a global scale are still scarce. Popp et al.[24] estimated the amount of urban land area under the SSP narratives across different regions, but the study lacked spatial details. Other global spatial projections typically have coarse spatial resolutions between 5 arc minutes and 0.5°[25,26]. In the latest CMIP6 (Coupled Model Intercomparison Project Phase 6) Land Use Harmonization dataset (LUH2), urban land is represented fractionally, with a resolution of only 0.25°[27].

The lack of sufficient spatial details in the existing SSP scenario projections may create uncertainties in environmental impact assessments[28]. A recent study has warned that the common land cover product could produce major distortions in land cover patterns at a 10-km resolution[9]. Our projections have a fine spatial resolution of 1 km, which preserves spatial details and can avoid the distortions in global urban land patterns. To the best of our knowledge, our results are the world's first 1-km resolution maps of future global urban land under the SSP framework.

The technical implementation of our projections is based on the Future Land-Use Simulation (FLUS) model[29]. This model utilises a machine learning approach to capture the complex relationships between urban land expansion and its driving factors. This model also adopts the mechanisms of cellular automata (CA)[29–31], which are capable of reflecting the complexities of path-dependence and positive feedback in the actual processes of urban land expansion[32].

In this paper, we project the future urban land expansions at 1-km resolution under SSPs by the FLUS model, and explore the potential impacts of the expansions in three aspects: first, the pressure of urban population decline; second, encroachment on cropland and natural habitats; third, the risk of producing major food crops.

## Results

**Future urban land demand.** We found substantial differences in the projected paths of future urban development across the five scenarios (Fig. 1a). Scenario SSP5 yields a monotonously increasing trend and the greatest urban land areas. Scenarios SSP2 and SSP3 yield the trends similar to that in SSP5, albeit with much smaller estimated urban land areas. For scenarios SSP1 and SSP4, a turning point is consistently observed in the 2080s and 2070s, respectively, after which urban land demand is expected to decline due to the assumed slowdown in socio-economic growth.

We also identify large disparities in the urban land demand projections among different macro regions. We select three representative macro regions, namely, China, the USA and LAM-L (i.e., low-income countries of Latin America), to demonstrate their distinctive development paths in the future (Fig. 1b–d, respectively). For China, which is currently the world's most populous country, its urban land demand is expected to increase rapidly before the 2040s or 2050s and decrease sharply thereafter in all scenarios. For the USA, divergent paths in future urban development are observed across all scenarios, in which SSP1, SSP2 and SSP5 consistently yield upward trends in urban land demand, while SSP3 and SSP4 predict downward trends after the 2050s and 2080s, respectively. The trajectories of future urban development in LAM-L, however, are in striking contrast compared to those observed in China and the USA. For LAM-L, scenarios SSP3 and SSP4 yield the largest urban land demand with linearly increasing trends, while scenarios SSP1 and SSP5 predict the smallest urban land demand. For all three regions, the uncertainty in the estimated urban land demand are the largest in scenario SSP5. Additionally, for LAM-L, the uncertainty is also large in SSP4 (Fig. 1d). Compared to the other two regions, the USA has the least uncertainty because it achieves a better fit in the panel data regression.

**Spatial distribution of future urban land expansion.** We simulate future urban land expansion at a spatial resolution of 1 km from 2020 to 2100 at 10-year intervals, with the year 2015 as the starting point. Details of model performance are shown in the supplementary information file (Supplementary Tables 1 and 2). Figure 2 shows the simulated urban areas in 2100 in scenarios SSP3 and SSP5 for the selected representative regions of China, the USA and LAM-L (the final global products for each SSP scenario are shown in Supplementary Figs. 1–5). As the coverage of urban areas is substantially smaller compared to those of other land covers, to improve the presentation in this paper, we implement a focal summation analysis, with a radius of 15 grids, on the results of the simulated urban land maps (Fig. 2). China and the USA, as two representative countries for middle-income

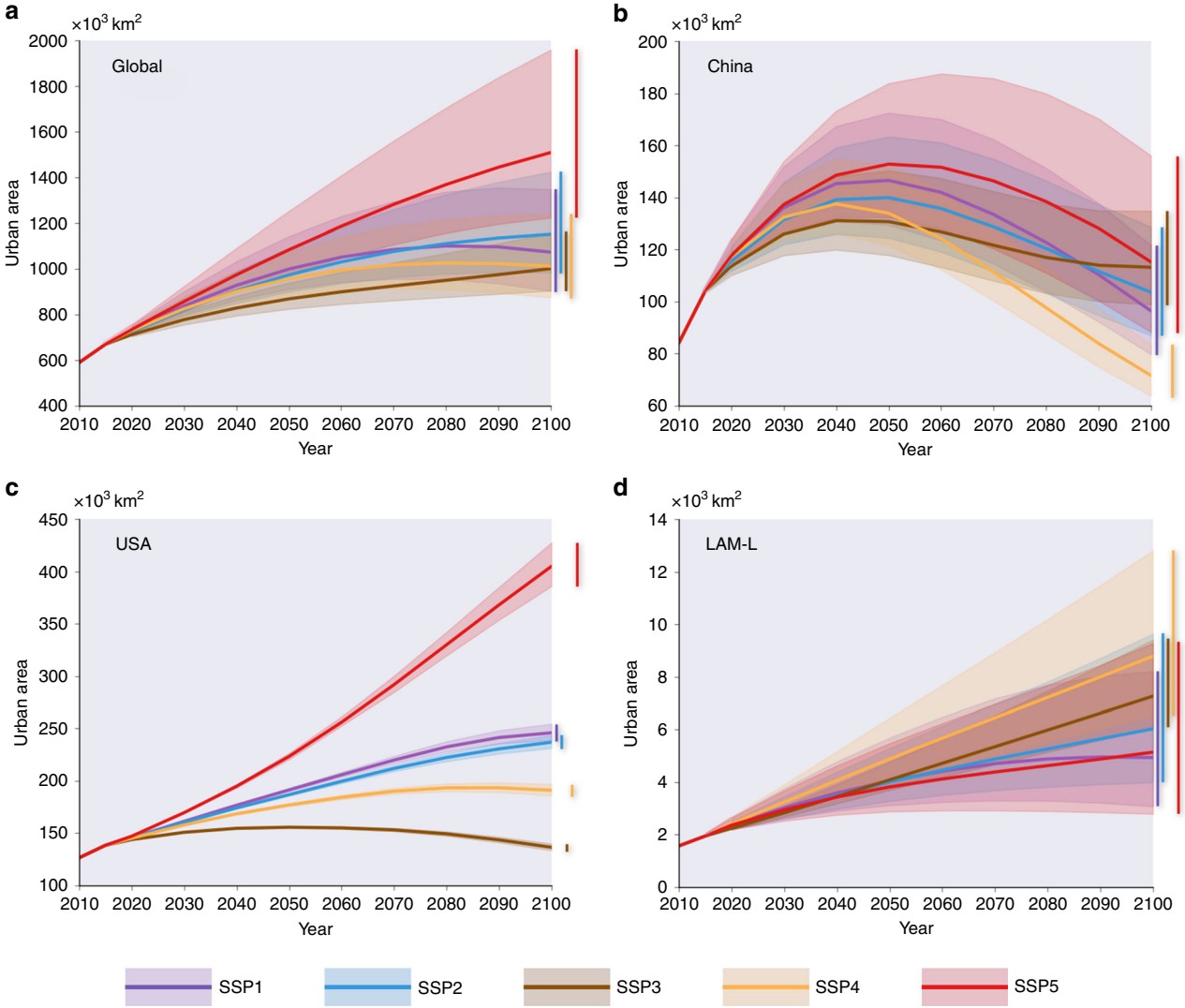

**Fig. 1 Projections of urban land demand globally and in the representative macro regions for 2010–2100 under the SSP scenarios. a** Global, **b** China, **c** USA, **d** the region LAM-L consists of the low-income countries of Latin America, including Belize, Guatemala, Haiti, Honduras and Nicaragua. The shaded areas represent the 95% confidence intervals of the projected urban land demand.

and high-income levels, respectively, tend to have more urban land expansion in scenario SSP5 than in SSP3. However, the LAM-L region, which mainly consists of low-income countries, experiences more urban land expansion in scenario SSP3 than in SSP5. Most of the newly expanded urban land in all of these countries is expected to take place at the edges of the existing highly urbanised areas.

To evaluate the spatial uncertainty in the simulation results, we performed 100 simulations for each scenario during the future scenario simulation. Then, overlay analyses of the simulation results were conducted to determine the probability of a city appearing at each location in space. Figure 3 shows the spatial probability distribution of urban growth in the SSPs in three major metropolitan areas by 2100, i.e., the spatial uncertainty in the simulation results. Compared with the 2000–2030 global urban simulation at a 5-km resolution by Seto et al.[8], our results show a similar urban spatial pattern, such as in the North China Plain and the Yangtze River Delta region, which are regions with rapid urban land growth.

The implementation of a patch-based simulation strategy has successfully revealed the changes in urban systems in terms of the rank-size distributions of urban land patches (i.e., $ln(rank_{patch}) =$

$\lambda \ ln(size_{patch}))$ (Fig. 4). For China, despite the increased total urban land areas, the $\lambda$ values change slightly from $-1.24$ in 1992 to $-1.22$ in 2015. Similarly, for the USA, the $\lambda$ values are $-0.99$ and $-0.96$ in 1992 and 2015, respectively. In future scenarios, the $\lambda$ values for China and the USA continue to increase, while the $\lambda$ values for the LAM-L region are expected to increase from $-0.91$ in 2015 to $-0.83$ (SSP3) and $-0.89$ (SSP5) in 2100, suggesting a relatively rapid increase in small urban land patches in LAM-L.

Our projections are comparable to existing global urban land projections, except that our results have a much higher spatial resolution. We choose three representative products for the comparison. The first one is the 2030 global urban expansion product based on a single UN scenario at a 5-km resolution, which is created by Seto et al.[8]. The second one is the 2050 global urban growth projection based on a historical trajectory with a 1-km resolution, which is created by Zhou et al.[12]. The third one is the 0.25-degree (~27-km on the equator) LUH2 dataset that follows the assumptions of SSPs[27]. As there is no historical trajectory in the SSPs, we select the results of the middle pathway (SSP2) to compare with the single scenario projections mentioned above. The comparison is implemented in different regions, as shown in Fig. 5. The distribution of urban land areas is similar

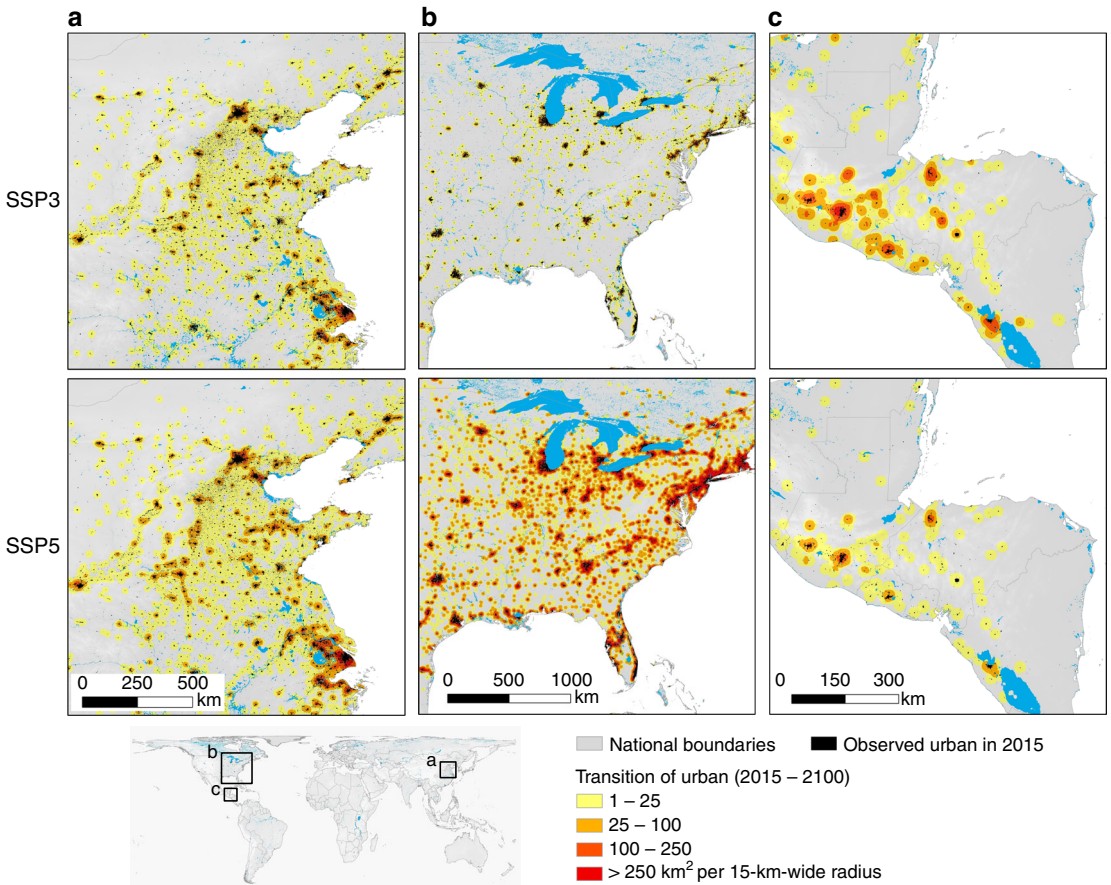

**Fig. 2 The simulated urban land maps in 2100 for representative regions in scenarios SSP3 and SSP5. a** China, **b** the USA and **c** LAM-L. Focal statistics have been applied to the simulated maps for better visualisation. The results show that China and the USA experience more urban land expansion in scenario SSP5 than in SSP3. In contrast, the LAM-L region experiences greater urban land expansion in scenario SSP3 than in SSP5.

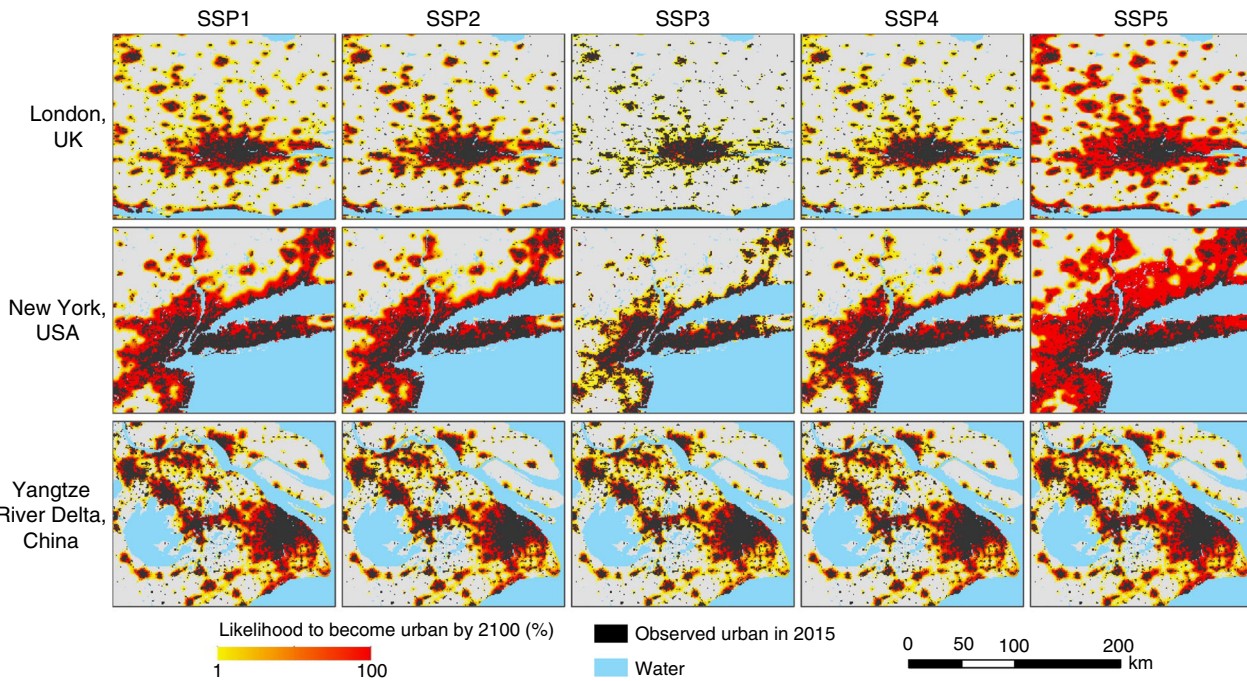

**Fig. 3 Uncertainty in the simulation results of three major metropolitan areas in 2100.** This figure shows the likelihood of each grid to become urban, which is estimated by overlapping the results of 100 simulation runs. Grids with higher agreements of simulated growth among the 100 simulation runs are assumed more likely to become urbanised.

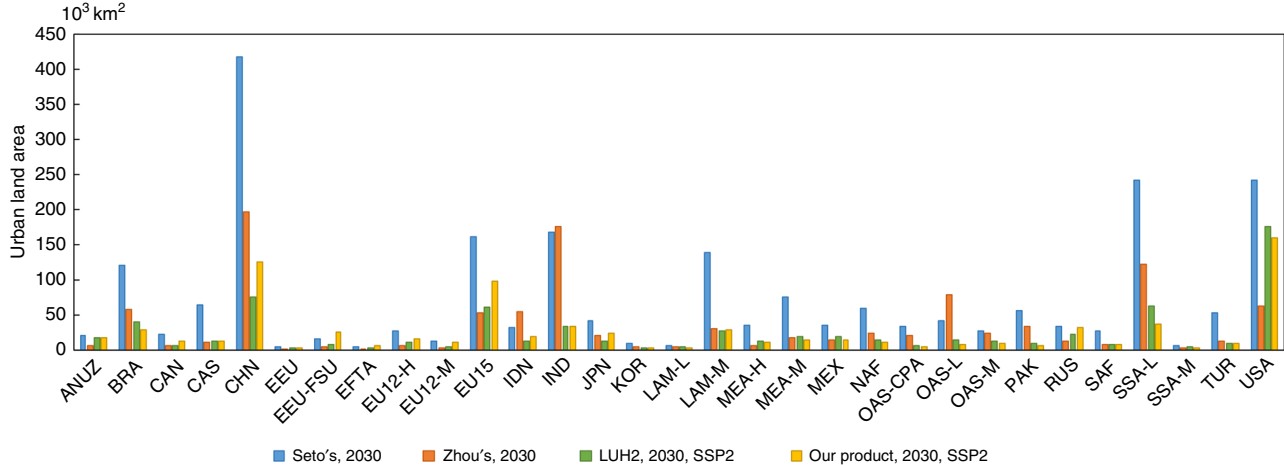

**Fig. 4 The rank-size distributions of urban land patches in the representative regions in different periods.** The implementation of a patch-based simulation strategy has successfully revealed the changes in urban systems in terms of the rank-size distributions of urban land patches. The LAM-L region consists of the low-income countries of Latin America, including Belize, Guatemala, Haiti, Honduras and Nicaragua.

**Fig. 5 Comparison of urban land area of 2030 in different regions by Seto's, Zhou's, LUH2 and our model.** Our results are significantly correlated with the three other results. Our results are most consistent with the urban land areas in LUH2 by a significant Pearson correlation coefficient of 0.93. Our results are also significantly correlated with Seto's and Zhou's by the coefficients of 0.82 and 0.56, separately. The abbreviations of the regions are defined following the official SSP dataset (https://tntcat.iiasa.ac.at/SspDb).

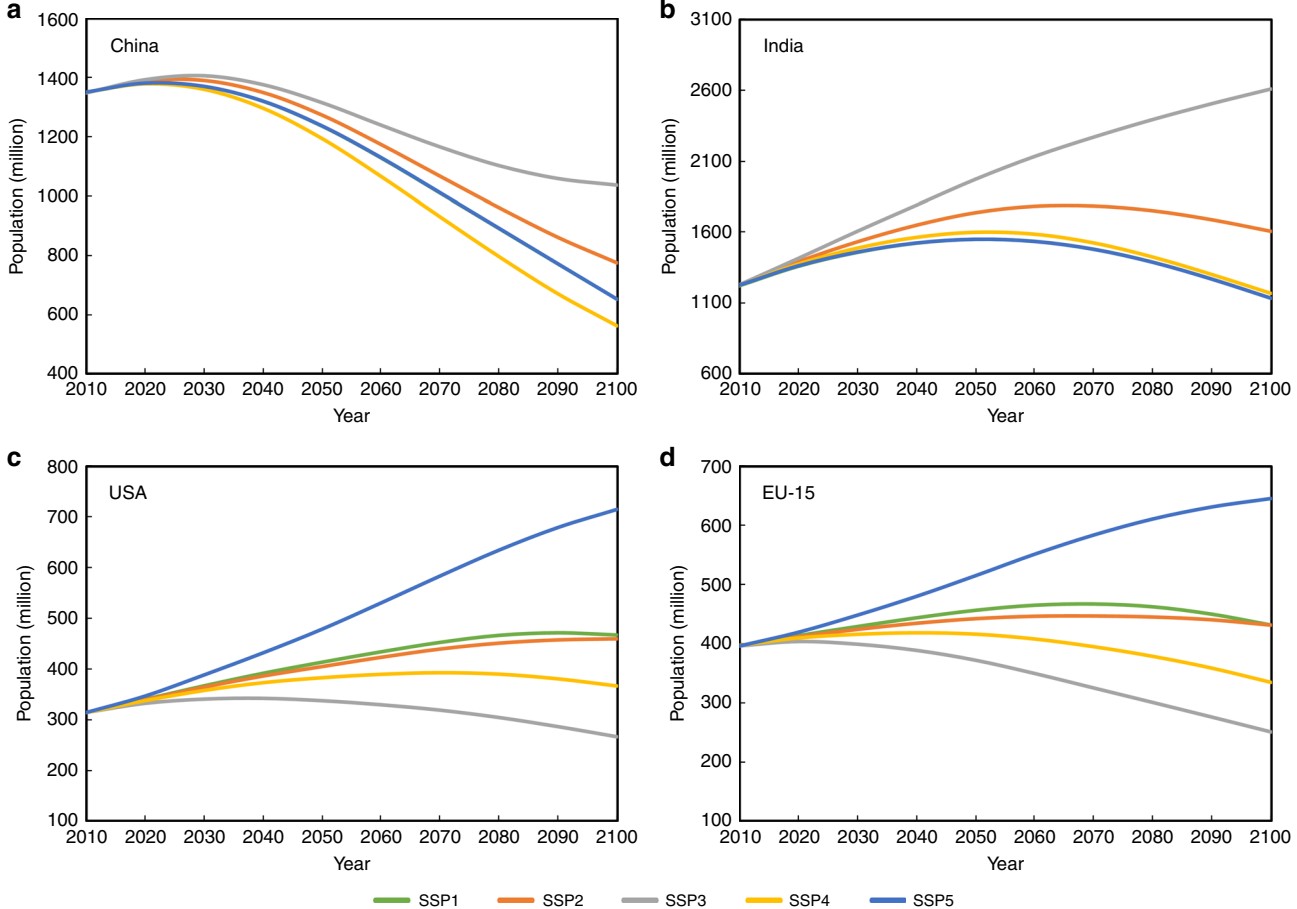

**Fig. 6 Population projections in the representative regions facing the pressure of urban population decline from 2010–2100 under the SSPs.** In China (**a**) and India (**b**), the population projections for the SSP1 and SSP5 scenarios are so close that their curves appear to be completely overlapping. EU-15 represents the European Union member states that joined prior to 2004, including Austria, Belgium, Denmark, Finland, France, Germany, Greece, Ireland, Italy, Luxembourg, Netherlands, Portugal, Spain, Sweden and the United Kingdom.

among the selected projections. Our results are most consistent with the urban land areas in LUH2, as indicated by a significant Pearson correlation coefficient of 0.93. The consistency is also high between our projections and Seto's results (Pearson correlation coefficient = 0.82). Our results show partial agreements with Zhou's results, as indicated by a relatively lower but significant Pearson correlation coefficient of 0.56. The spatial agreements between our projections and the three selected products are shown in Supplementary Fig. 6. Some evident spatial differences between our results and Seto's/Zhou's results can be found in regions such as the North China Plain. It is mainly because their projections do not consider the population decline trend in these regions, which can cause urban land growth to become stagnant[33].

**Pressure of future urban population decline.** Noticeable declines in urban land demand are observed in our scenario projections, mainly owing to the decrease in population. A typical example is China (Fig. 1b), in which urban land demand sharply decreases in all scenarios after the 2040s/2050s due to the declining population (Fig. 6). Urban population decline does not necessarily lead to a massive land conversion from urban to non-urban areas, although some individual cases of green recovery from urban land can be found in contemporary Germany at the block scale[34]. Rather, one possible consequence of urban population decline is the abandonments of the already built-up areas.

This has been occurring in some countries around world[35,36], even in the rapid urbanising country of China[37].

However, to address the decline in urban land demand in the spatial simulations, we assume that the land conversion of non-urban land to urban land is irreversible and that the massive conversion of urban land to non-urban land is not allowed. For a certain region, if its estimated urban land demand is smaller than its total area of the already built-up, then no changes are simulated for this region, and the spatial extent of existing urban land also remains unchanged. Subsequently, for regions experiencing a decline in urban land demand, we assess their pressure on urban population decline, which is defined as the percentage of the area of urban land surplus (unnecessary urban land compared to urban population) over the area of existing urban land, ranging from 0 to 100%. The results (Fig. 7) suggest that China will no doubt face the most severe pressure of urban population decline compared to other countries around the world. South America and almost all Asian areas will encounter substantial pressure of urban population decline in the SSP1 and SSP4 scenarios, while European countries and North America will face increased pressure of urban population decline in the SSP3 scenario.

**Structures of land from future urban developments.** We identify the land structures taken by future urban developments based on the Climate Change Initiative-Land Cover (CCI-LC) product for the 2015 baseline year. The results (Table 1) reveal that by

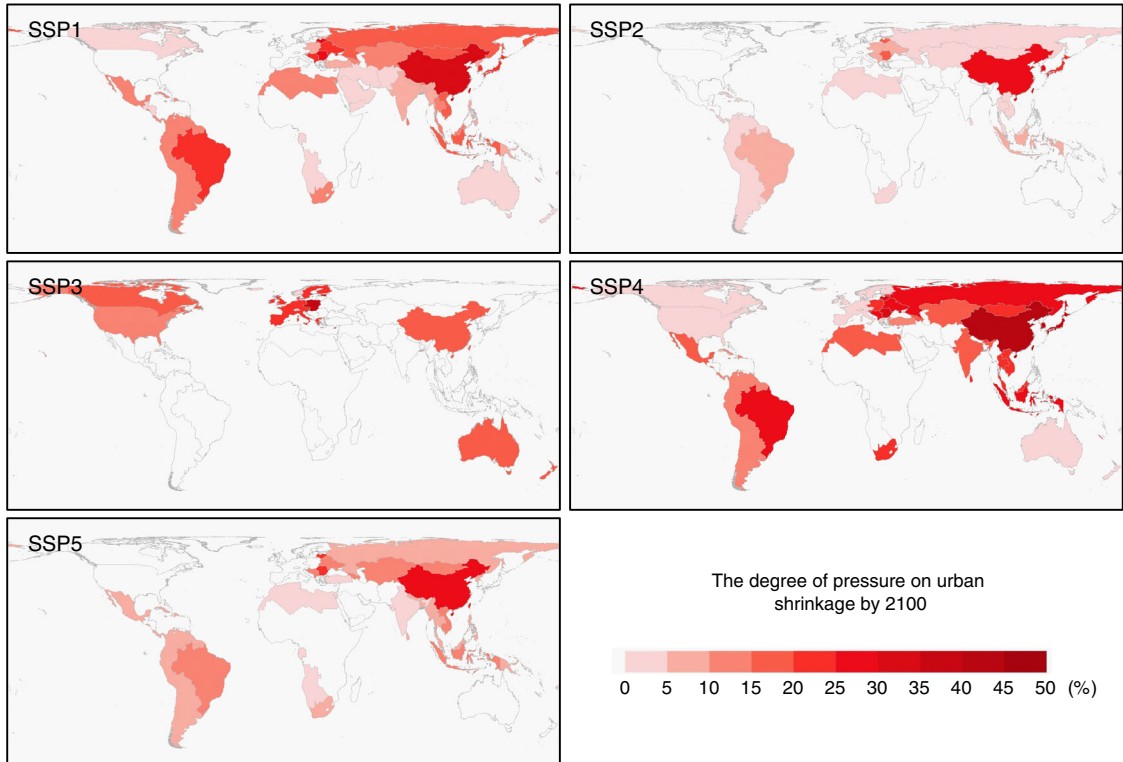

**Fig. 7 The pressure of urban population decline by 2100.** This figure shows the pressure of urban population decline by 2100 under five SSP scenarios. This pressure is calculated as the percentage of the difference between the existing urban land area and the estimated urban land demand to the existing urban land area.

**Table 1 Contributions of various land covers to newly developed global urban land by 2100 under the SSPs scenarios.**

|  | Proportion of the converted area to urban land (%) | | | | |
|---|---|---|---|---|---|
|  | SSP1 | SSP2 | SSP3 | SSP4 | SSP5 |
| Forest | 26.98 | 26.36 | 21.42 | 22.89 | 29.29 |
| Grassland | 12.01 | 10.86 | 5.71 | 7.49 | 14.64 |
| Wetland | 2.10 | 2.46 | 2.77 | 3.51 | 2.23 |
| Bare land | 3.76 | 4.68 | 7.28 | 5.86 | 2.98 |
| Cropland | 55.14 | 55.63 | 62.82 | 60.25 | 50.84 |
| Other | 0.01 | 0.01 | 0.00 | 0.00 | 0.02 |

2100, 51–63% of the newly expanded urban land will be converted from cropland. Urban encroachments on cropland will mainly occur in China, India, sub-Saharan Africa and western Europe (Supplementary Fig. 7). Forest and grassland collectively contribute to 30–44% of the newly expanded global urban land. Losses in forests due to urban land are expected to occur mainly in North America, South America, western Europe, sub-Saharan Africa and Australia (Supplementary Fig. 8). Major grassland declines due to urban land expansion will occur in the USA and western Europe (Supplementary Fig. 9). Global wetlands, which are one of the most biologically diverse ecosystems, will be consumed by 2–4% of the newly developed global urban land. Most wetland losses will take place in the USA and sub-Saharan Africa (Supplementary Fig. 10). Major bare land declines due to urban land expansion will occur in the Middle East (Supplementary Fig. 11).

**Impacts of urban expansion on food production.** Given that future urban expansion occurs mostly over croplands, we further assess the impacts of urban expansion on food production. We estimate the production losses caused by urban expansion for rice, wheat, maize, potato and vegetables using the crop maps developed by You et al.[38]. The results (Fig. 8) indicate that the scenarios SSP3 and SSP4 yield the smallest amount of estimated crop production losses, mainly because of the decrease in socio-economic developments and urban expansion processes. In contrast, scenario SSP5 reports the greatest amount of crop production losses due to the assumed rapid urban expansion. Overall, by 2100, global urban land expansion will cause declines of 2–3% in rice production, 1–3% in wheat production, 1–4% in maize production, 1–3% in potato production and 2–4% in vegetable production (see Supplementary Figs. 12–16 for more spatial details). The estimated number of people affected by the losses of crop production vary widely from 122 million to 1389 million, depending on which food consumption statistics are used (Table 2). Note that these results are not the numbers of people suffering from starving, but the maximum numbers of people affected by the production losses of a single food crop type, if these losses cannot be compensated with productions of other food crops. In addition, we did not consider yield changes due to changes in agricultural management practices which might compensate the projected losses.

While urban areas are located predominantly on productive land, urban developments often prevail in direct competition with crop production[39]. Global urban expansion can, therefore, exert important impacts on crop production that are beyond the losses of existing croplands. Our projections (Fig. 8) demonstrate that future urban land expansion will cause up to 1% of loss in the global cropland area. However, the direct losses of global crop production are likely to exceed 1% and can even reach as high as 4%, depending on the specific crop types. These results suggest the importance of conserving high-quality croplands to reduce the impacts of urban expansion on food production capacity.

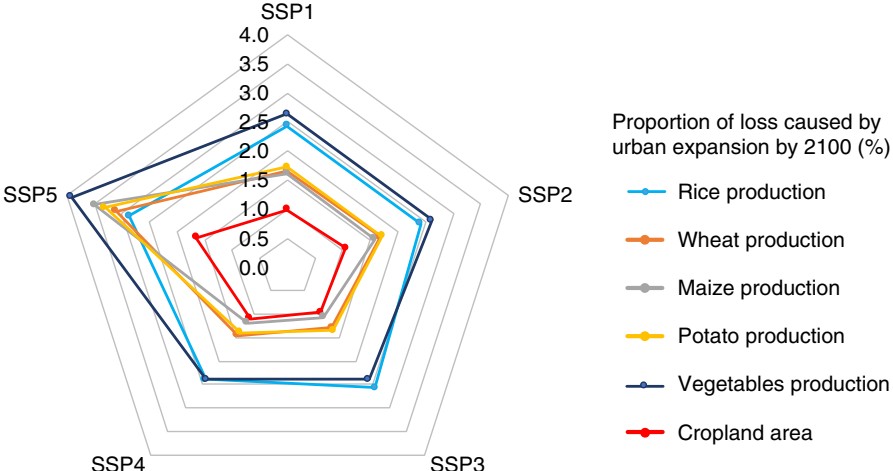

**Fig. 8 Estimated cropland and food production losses caused by urban expansion by 2100.** Comparing to the approximately 1% loss in the global cropland area caused by future urban land expansion, the direct losses of global crop production are likely to exceed 1%, and can even reach as high as 4%, depending on the specific crop types.

**Table 2 Global food production losses caused by urban expansion by 2100.**

| | Global food production losses caused by urban expansion ($10^6$ ton) | | | | | Population affected by food production losses (million person) | | | | |
|---|---|---|---|---|---|---|---|---|---|---|
| | SSP1 | SSP2 | SSP3 | SSP4 | SSP5 | SSP1 | SSP2 | SSP3 | SSP4 | SSP5 |
| Rice | 15.19 | 15.09 | 16.04 | 14.99 | 17.88 | 281.77 | 279.86 | 297.51 | 278.03 | 331.62 |
| Wheat | 10.31 | 10.35 | 7.99 | 8.98 | 19.19 | 157.50 | 158.25 | 122.12 | 137.22 | 293.35 |
| Maize | 11.53 | 11.21 | 7.56 | 8.41 | 24.84 | 644.70 | 626.72 | 422.86 | 470.19 | 1388.70 |
| Potatoes | 5.46 | 5.47 | 4.29 | 4.42 | 10.50 | 159.91 | 160.09 | 125.48 | 129.36 | 307.31 |
| Vegetables | 19.99 | 19.84 | 18.17 | 18.06 | 29.67 | 183.55 | 182.18 | 166.79 | 165.79 | 272.44 |

We estimated the population affected by food production losses according to the latest annual per capita food needs derived from the Food and Agriculture Organization (FAO) of the United Nations (http://www.fao.org/faostat/en/#data).

## Discussion

In this paper, we present the long-term scenario projections of future urban land expansion with a fine spatial resolution of 1 km. Moreover, our dataset is based on the latest SSP scenarios under CMIP6. This can enhance the usefulness of our projections to support the research in other related disciplines, such as ecological protection[40], water security[1], urban climate[41] and global climate change[4].

The uncertainty of modelling can be divided into stochastic uncertainty, parameter uncertainty, heterogeneity and structural uncertainty[42]. We quantified the parameter uncertainties in the projections of urban land demand (Supplementary Table 3). In order to deal with the heterogeneity, we assigned different intercepts to different regions in the panel data regression (Supplementary Table 3), implementing spatial simulations in different regions separately. We then performed 100 spatial simulations to understand the stochastic uncertainty of the model. To address the structural uncertainty[42], we compared our products with three other products using different models (Fig. 5 and Supplementary Fig. 6).

In addition to the regional-scale urban expansion projections as analysed in the previous section, we also analysed the urban expansion trends of the three typical international metropolitan areas under different SSPs, including New York, London and the Yangtze River Delta (Supplementary Fig. 17). Although these three metropolitan areas are in countries with different development stages, their urban expansion trends are similar: by 2100, the urban

area will become the largest in SSP5, followed by SSP1, SSP2, SSP4, and SSP3. The difference is that the urban expansion trends of New York and London in SSP2 are closer to those in SSP1, but the trend of the Yangtze River Delta in SSP2 is closer to that in SSP4. Moreover, the urban expansion in these selected metropolitan areas is evidently faster than that of their countries' average, except those in the developed countries in the SSP5 scenario (Supplementary Table 4). Therefore, rapid urban expansion may further increase the pressure on the environments and resources of these metropolitan areas which are of dense population.

Based on the urban land projections, we have identified the pressure caused by future urban population decline for different regions across the world. Urban population decline occurrences largely correspond to low birth rates and population ageing. Typical regions that will suffer from substantial pressure from urban population decline mainly include China and many other Asian countries. North America, South America and western Europe experience great pressure from urban population decline mainly in scenarios SSP3 and SSP4. These results have important implications for urban planning and management. In rapidly urbanising countries, such as China, urban planning is usually prepared with the expectation of growth, with little or no awareness of the ongoing/upcoming challenges caused by urban population decline[37]. A direct consequence of urban population decline is the difficulty of sustaining the maintenance of infrastructure, which was originally built to serve a larger population, leading to declines in livelihoods and quality of life in cities.

Abandonments and social disorder are also potential outcomes raised by urban population decline that threaten public health and safety in cities[43,44]. Moreover, from an economic point of view, urban population decline may cause economic recession because of the negative impacts on employment, industrial production and property value[45–47].

Despite the potential decline of urban population in some regions of the world, global urban land is expected to experience rapid growth before the 2040s. Our projections have demonstrated that most future urban land expansion occurs over croplands (50–63%) and forests (30–44%), causing the losses of food production and supply. To offset these losses, however, reclaiming cropland remains as a most immediate way, because land productivity may be difficult to improve substantially in the foreseeing future. The losses in croplands and forests will also have profound impacts on ecosystem services, such as carbon sequestration, habitat provision and food supply to human societies[48–51].

Note that in this study we only account the direct impacts of urban expansion on other land types. To gain a clearer understanding, we complement our results with those obtained from Popp et al.[24], which provide scenario projections of cropland expansion on other land use types. Their results suggest that the SSP3 scenario features the largest amount of cropland expansion (~700 million ha from 2005 to 2100), while the SSP1 scenario has the smallest increase of cropland area (~20 million ha from 2005 to 2100). For SSP2, SSP4 and SSP5, the cropland areas are expected to increase between 250 million ha and 400 million ha from 2005 to 2100. Our scenario projections, which mainly focus on the direct impacts of urban expansion on cropland, reveal that SSP5 has the largest cropland loss due to urban growth (37.6 million ha), while SSP3 has the smallest loss (22.0 million ha). The estimated cropland losses in other SSP scenarios vary slightly from 22.4 million ha to 24.9 million ha. Therefore, the combination of our results and Popp's reveals the contrasting patterns of urban land and cropland expansion. In the scenarios with more cropland losses caused by urban expansion, there are relatively small increases of cropland, and vice versa.

Our results show that the crop production losses induced by urban expansion will affect the annual needs for a certain food of ~122–1389 million people. More importantly, the crop production losses (~1–4%) are disproportional to the cropland land losses (~1%). Previous studies have revealed that a 2-°C global temperature increase will cause 8–14% of global maize production losses[52]. In comparison, our projections suggest that urban land expansion alone will lead to 1–3% of global maize production loss, indicating the profound impacts of urban expansion on global crop production. These results imply the critical importance of conserving fertile and productive cropland. Governance and proper planning of urban land development locations and procedures is a key measure to reduce future losses in crop production[39,53]. However, the governance qualities appear to be moderate to low due to the fragmented governance structures in many countries that will suffer from large crop production losses[54]. The effective protection of croplands, therefore, calls for increasing efforts in educating policy makers and the development of more comprehensive governance regimes[55].

There are several limitations in our projections. First, we do not address the potential errors in future urban land expansion that are caused by the misclassifications in historical urban land maps, which were acquired from the GHSL dataset. Second, we hold the spatial driving factors of urban land expansion constant in the scenario projections, which is mainly because of the difficulty in the reliable prediction and uncertainty of, for example, changes in urban infrastructure. Third, urban development policies are usually operated at a country level, but in this study, we provide urban land demands at a regional scale, mainly due to the lack of country-level scenario data. We will try to solve this limitation in our future work by complementing more data to obtain the relevant country-level information for different scenarios. Forth, we do not consider the impacts of future climate change on land-use change. This limitation can be alleviated in our future work by applying the projections under the integrated scenarios of RCP-SSP, which take into account the feedbacks of global climate changes.

## Methods

**Study design.** We use panel data regression to estimate future urban land areas based on the factors of population, urbanisation rate (percentage of urban population to total population) and gross domestic product (GDP). Here panel data regression refers to a regression method that simultaneously uses cross-sectional and longitudinal data. We follow the region definitions in the SSP database (https://tntcat.iiasa.ac.at/SspDb), which aggregate the world's countries into 32 macro regions according to the conditions of geographic location and income level (i.e., high-income, mid-income and low-income levels). The historical urban land areas for the years 1975, 1990, 2000 and 2014 are acquired from the Global Human Settlement Layer (GHSL) dataset[56] (Here urban land refers to artificial cover and paved surfaces). The concurrent statistical data of population, urbanisation rate and GDP are obtained from the World Bank[57] and United Nations[58]. We then apply the built panel data regression model to project the urban land areas for each scenario based on the predictions of future population, urbanisation rate and GDP, which are available in the SSP database.

We simulate the spatial distribution of urban land expansion using the FLUS model. This model employs artificial neural networks (ANNs) to estimate urban development potential using a set of spatial driving factors (e.g., population, GDP, distance to city centre, distance to road network, distance to airport, elevation, slope, eco-region and water resource condition). The spatial simulation of future urban development, as constrained by the estimated potential, is based on CA. We further enhance the performance of the FLUS model by incorporating a patch-based simulation strategy[59–61], aiming at preserving the scaling property of actual urban systems[61,62] (a detailed flowchart is presented in Supplementary Fig. 18).

**Estimation of urban land demand.** We use the panel data regression model to estimate the urban land demand for each SSP scenario. Specifically, we use historical statistical data and urban land data to establish the relationships between per capita urban land demand (CA′) and the explanatory variables of per capita GDP (GDPC) and urbanisation rate (PU) at the macro regional level:

$$\mathrm{CA}'_{r,t} = \beta_0 + \beta_1 \times \mathrm{GDPC}_{r,t} + \beta_2 \times \mathrm{PU}_{r,t} + \sum_{i=2}^{N} \alpha_i \times Z_{r,i} + \varepsilon_{r,t} \tag{1}$$

$$r = 1, \dots N$$

where $Z$ is the regional dummy variable; $\varepsilon$ is the error term; and $r$ and $t$ refer to the region and year, respectively.

A total of 32 macro regions are considered (the extent of each region is shown in Supplementary Fig. 19) according to the regionalisation scheme of the SSP database (https://tntcat.iiasa.ac.at/SspDb). The historical urban land data are acquired from the GHSL dataset[56], which cover the years 1975, 1990, 2000 and 2014. The concurrent statistical data of population, per capita GDP and urbanisation rate are collected from the World Bank[57] and United Nations[58] and aggregated at the macro regional level. The estimated coefficients have passed a significance level of 0.01. The values of $\beta_0$ in each region reflect the heterogeneity between different regions[42] (detailed results of the estimated coefficients are shown in Supplementary Table 3).

After it is established, Eq. (1) can then be applied to the prediction of future per capita urban land demand using the scenario projections of per capita GDP and the urbanisation rate provided by the SSPs database (https://tntcat.iiasa.ac.at/SspDb). The regional urban land demand in a future year, $t+$, therefore, can be calculated by multiplying the estimated per capita urban land demand at year $t+$ by the projected total regional population at year $t+$. The global and selected regional results of future urban land demand, with 95% confidence intervals, are shown in Fig. 1.

**Spatial simulation of urban land expansion and validation.** We use the FLUS model to simulate future urban land expansion at a 1-km resolution. This model adopts several assumptions: first, urban conversion on other land cover types is irreversible and hence any recovery from urban land to non-urban land is not allowed; second, for any regions, if the estimated urban land demand is smaller than the existing area of land that is already built-up, then no changes are simulated; third, the spatial driving factors are held constant in future projections of urban land expansion.

The FLUS model is established under the framework of CA, which has been widely applied in land change modelling[9,29]. This framework assumes that the probability of a non-urban grid being converted into an urban grid (TP) is a product of the urban development potential (P; also termed the probability-of-occurrence), neighbourhood effect ($\Omega$), development restriction (con) and adjustment factor (inertia).

The urban development potential (P) represents the site conditions for urban development according to a set of spatial driving factors (see Supplementary Table 5 for the full list of these factors). Specifically, for each macro region, urban and non-urban grid samples in a balance are collected and used to train an ANN classifier to yield the urban classification probability, with the spatial driving factors as input features. The trained ANN classifiers are then used to estimate the urban classification probability for each grid throughout the entire region. The resulting urban classification probability is regarded as the urban development potential (P). Note that this procedure is implemented separately for each of the 32 macro regions.

The neighbourhood effect ($\Omega$) factor is calculated as the fraction of existing urban land area in a neighbourhood consisting of $5 \times 5$ grids. This factor represents the effect of positive feedback during actual urban development, i.e., new developments are more likely to occur in/near places that are already built-up. The development restriction (con) factor has binary values, with 0 referring to the condition of no developments (otherwise, 1). The adjustment factor (inertia) is used to adjust the growth rates of urban land in the simulations and facilitate convergence towards an expected quantity.

To preserve the scaling property of actual urban systems, we further adjust the conversion probability (TP) using a patch-based urban growth strategy. The scaling property of actual urban systems refers to the characteristic of far more small things than large ones in urban systems[63]. This property has been identified in the distribution of cities around the world[61] and the distribution of urban land patches at the intra-urban scale as well[62]. The scaling property of urban systems is the result of, and is enhanced by, the rich get richer effect, which means that large entities (e.g., cities or land patches) grow faster than small ones. The patch-based urban growth strategy, therefore, is to represent this effect when shaping urban systems. Specifically, the conversion probability (TP) of a non-urban grid is increased if it is located near a large patch. The larger the patch that this non-urban grid is close to, the greater the increase in the conversion probability (denoted as $TP_{patch}$) (the technical details of this adjustment are described in the Supplementary Information).

The land conversion from non-urban to urban land is then simulated following these procedures (i.e., the roulette selection): first, randomly select a non-urban grid and compare its adjusted conversion probability $TP_{patch}$ against a randomly generated value of [0, 1]; second, convert this non-urban grid into an urban grid if $TP_{patch}$ is greater than the random value; otherwise, the grid remains unchanged; and third, repeat the first and second steps until the simulated urban land expansion satisfies the expected quantity of the new urban land.

The agreement between the simulated and observed urban land expansion is evaluated using the FoM (Figure of Merit) indicator[64]. We use this indicator because it avoids the drawback of accuracy overestimation in conventional validation metrics (e.g., the Kappa coefficient)[65]. FoM is calculated as the ratio of the correct predicted change to the sum of the observed change and predicted change. The value of FoM ranges from 0 to 1, with 1 representing a perfect fit between the simulated and observed changes. However, existing applications of land change modelling usually report FoM values of 0.1–0.3[9,60] because of the path-dependence effects[32] that hamper land change models in making more accurate predictions. Nevertheless, the performance of our model is reliable, as the resulting FoM values are similar or greater than those of applications with land change modelling at a global scale[9]. Please refer to the Supplementary Information for more details on the implementation and validation of the FLUS model.

In the predictions of future urban development, due to the projected downward demographic trends, certain regions are likely to encounter the declines in urban land demand (e.g., China), which may further reach a level that is below the regional total of existing urban land. That is, the existing urban land is sufficient in satisfying the needs of the shrinking population. In this case, no changes in urban land are simulated, and the spatial extent of existing urban land also remains unchanged.

**Estimation of food production losses and affected population**. We estimate food production losses based on the global maps of crop production (SPAM 2005) developed by You et al.[38]. These maps contain 27 major categories of food crops, with a spatial resolution of 5-arc minutes (~10 km at the equator). As the crop production maps are much coarser than the 1-km resolution land-use data used, we assume that crop productions are the same within a 10-km grid, and therefore, the crop production losses are proportional to the areal losses in croplands within a 10-km grid. By adopting this assumption, we estimate the crop production losses according to the identified conversions from cropland to simulated urban land. We selected five major food crops (rice, wheat, maize, potatoes and vegetables) to estimate their production losses. The choice of these crop types is based on their important roles in global crop production (see Supplementary Fig. 20 for their respective proportions to the global crop production).

The diet structures of people vary largely from one place to another. To estimate the number of people affected by the global crop production losses, we use the global average per capita food supply of rice, wheat, maize, potatoes and vegetables as the basis for the estimation. These statistics can represent the overall situation of food supply worldwide and hence are adequate to apply in our analysis (http://www.fao.org/faostat/en/). These statistics cover a wide range of crop types over multiple years. Specifically, we use the 2013 statistics, which are the latest statistics. We then estimate the affected population by dividing the estimated losses of a certain crop type (e.g., rice, wheat, maize, potatoes and vegetables) by the global annual per capita consumption of that crop type. The results for the five SSP scenarios are provided in Table 2.

**Reporting summary**. Further information on research design is available in the Nature Research Reporting Summary linked to this article.

## Data availability

The historical GDP and population statistics are available from http://www.fao.org/economic/ess/ess-economic/gdpagriculture/en/. The historical proportion of the urban population can be retrieved from http://www.un.org/en/development/desa/publications/2014-revision-world-urbanization-prospects.html. The historical area of urban land cover for the panel data regression is calculated from the GHSL dataset, which is available at https://ec.europa.eu/jrc/en/scientific-tool/global-human-settlement-layer. The official SSP database can be obtained at https://tntcat.iiasa.ac.at/SspDb/. The CCI-LC product is available from http://maps.elie.ucl.ac.be/CCI/viewer/download.php. The spatial variables for estimating the probability-of-occurrence surfaces are collected from various sources, the list of which is available in Supplementary Table 5.

## Code availability

We used the FLUS software for generating modelling results in this manuscript. FLUS which is freely accessible to all users can be downloaded at http://www.geosimulation.cn/flus.html. Specific FLUS configurations and modifications to be used to generate the results in this study are described in the Supplementary Information.

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

## Acknowledgements

This study was supported by the Key National Natural Science Foundation of China (Grant No. 41531176), the National Key R&D Program of China (Grant No. 2017YFA0604402), the National Natural Science Foundation of China (No. 41871306), and the Southern Marine Science and Engineering Guangdong Laboratory (Zhuhai) No:99147-42080011.

## Author contributions

G.C. and X.Li contributed equally to this work; X.Li and X. Liu designed the research; G.C. performed experiments and computational analysis; G.C. and X.Li drafted the manuscript; G.C., X. Li, X. Liu and Y.C. amended the manuscript; X.Liang and K.H. provided technical support; G.C., Y.C., X.Xu and W.L. improved the design of experiments; J.L., Y.Q. and Q.W. contributed to the preparation of the experiments and computational analysis.

## Competing interests

The authors declare no competing interests.
