## [Peer Review File · Nature Communications]

Reviewers' comments:

Reviewer #1 (Remarks to the Author):

General thoughts:

This paper deals with an important topic (urban growth), and presents some novel model results. However, I have several major concerns:

Major concerns:

1.) The paper's title suggests a focus on the environment impacts of urban growth, but the bulk of the paper is about the details of urban growth forecasting (e.g., Zipf's law or not, how much urban area in which region, etc.). If the goal of the paper is to present the details of urban growth forecasting, then make that the focus of the title and drop the brief mention of things like soil organic carbon. On the other hand, if this paper is trying to really focus on environment impacts, then they need much more coverage in the paper.

2.) This paper says that the 1km urban area forecasts were developed for this paper. This appears to be incorrect, since I believe (although I am not certain) these are the same data published in the Li et al. (2017) paper in ANNALS of THE AMERICAN ASSOCIATION OF GEOGRAPHERS. There's nothing wrong with using this dataset for this new analysis of urban growth impacts, but it should be clear (in the Abstract and elsewhere in the text) that these are the same 1km datasets that have been previously published on. For instance, in the main text it says "This study creates a global urban land-use simulation product", when it appears this was largely the same as the dataset published in the 2017 paper. The authors need to clarify if this is a new dataset, or the same as the one in the ANNALS paper.

3.) The authors present an impressive forecasting methodology for urban growth, which would allow them to estimate the uncertainty in their estimates, both during the panel regression stage (which would allow confidence intervals of urban area by region) and at the pixel-level stage (since pixels are shown as having a probability of being urban, there is presumably some estimate of uncertainty from the cellular automaton modelling). However, these uncertainties are not presented in the paper. If this paper is to be suitable for NATURE COMMUNICATIONS, then uncertain intervals for quantities like urban area need to be shown. To be clear, the authors do a good job of showing how the different SSPs influence urban area. I am not talking about the variation among SSPs, I am talking about the uncertainty in estimation of urban area within each SSP.

4.) The paper lacks a clear statement of the hypothesis that it is testing or question it is answering. Please state this clearly in the introduction section of the paper.

5.) The authors make a very strong assumption that urban population decline will result in urban area decline. This result seems to me to happen because they extrapolate their panel regressions (predicting per-capita urban land use at a point in time) to estimate change in urban area over time. This seems to me statistically questionable (see my comments below) and also at variance with the historical data (see my comments below about Manhattan, and about shrinking cities in the US and Europe). The authors do seem to try to mitigate this assumption with equation 6, but I am unclear how this was calibrated.

6.) The manuscript needs a more clear separation in the manuscript between methods and the results, where possible. Right now, substantial discussion of modelling methods is mixed in with the text presenting the model results.

Other comments about the manuscript:

I don't know the submission guidelines for this journal, but I would have preferred that there were

page and line numbers. Makes the tasks of reviewers easier.

Throughout the paper, suggest spelling out "China" rather than "CHN", as it is only two extra characters!

Title: The phrase "environmental effects" is rather broad, and can suggest all kinds of environmental considerations not focused on in this paper. I would suggest replacing with a phrase describing the specific environmental issues tested. As the paper is written, much of the discussion is about urban area and the area of other land uses it displaces (grassland, forest, etc.).

Abstract:

Overall, I worry this abstract will not be comprehensible to the average reader of this journal, although specialists in the field of urban growth modelling will understand it fine. I would suggest you remove some of the jargon with equivalent phrases to make clear to the average reader of this journal what is scientifically novel/newsworthy about this paper. The abstract really needs to stand on its own. Some specific suggestions:

The phrase "patch-based urban growth strategy" is not clear to the reader at this point. Please use another, more intuitive phrase here. Same is true for "polarized urban form". To a lesser extent, this is true for "Zipf's Law", although most readers in this discipline of urban growth will understand the sentence.

SP1 promotes better "human-nature harmony" in what sense? This feels very imprecise for a scientific abstract.

SOC is an abbreviation- follow journal usage as to whether this needs to be spelled out the first time you use it or not.

Introductory text:

Suggest "hives of human activity" be "centers of human activity".

Suggest that "is considered a trade-off between" be rewritten as "varies between".

This section needs a clear statement of the questions or hypotheses your work is testing.

Study design section:

"Adaptive inertia mechanism" and "roulette selection" are not clear to the reader at this point. If you want to use these phrases here, please define them briefly. Obviously, more detailed information can be found in the Methods section.

Magnitudes of urban land use section:

"takes a rocky road" is unclear and imprecise, please clarify what specifically about the SSP3 assumptions leads to lower urban growth. I know this is the shorthand name for SSP3, but a lot of readers won't, and it is more useful just to say what is happening with the assumptions of the scenario that is driving lower urban growth.

"However, all scenarios indicate that CHN will show a decrease in urban land use..." This result seems to me to be a result of some assumptions built in to how the panel regression of urban land use will be. Clearly, there is a correlation at a point in time between urban population and urban area, a quite predictable one you can build a regression model on. I even believe that the assumption of future urban growth being similar (e.g., at a similar urban population density) is plausible, although it is worth noting there's an assumption there when you go from historical data to forecasting. But I don't know if it plausible that a decline in urban population will lead to a decline in urban area. Are there good historical case studies of this? In Manhattan, for instance, there were more than twice as many residents in the 19th century as today, but urban area in

Manhattan hasn't declined significantly. Instead, urban population density has just fallen.

In America, we would write "fossil-fueled" rather than "fossil-fueled", but the later is fine in British English. Whatever is standard spelling for this journal is appropriate.

Please define "LAM-L" before use (list the countries, or give some other definition). Is it all the Latin American countries, or just those that are low income (excluding middle income countries like Brazil, Argentina, etc.)?

Spatial distribution of urban land use section:

Here, and elsewhere in the results sections, there are a few sentences about methods mixed in. For instance, there's a sentence about technical details of GeoSOS-FLUS. I would suggest having a more clear separation in the manuscript between methods and the results, where possible.

It is a simplification of Zipf's law to say that it "reflects the fact that larger urban patches grow faster". That may be the implication in this context, but Zipf's "law" (really just a statement of an empirically observed trend that has held constant over many decades) is really a statement about the city size distribution. If Zipf's law is going to be featured prominently in the results, then you should define what it means more fully in the Introduction section.

Model validation and simulation performance assessments section:

This section is very detailed. For a paper that was focused on modelling methods, this would seem appropriate. But it would be usual for a paper in NATURE COMMUNICATIONS to have this level of detail on model validation, etc., in the main text. That said, it is excellent you are testing different modelling methods using historical series of urban growth.

For "Figure of Merit", please give the reader a brief (1 sentence) description of how to interpret this metric. Does it range between 0 and 100%? What does it represent? Many readers without a background in remote sensing or statistics may not know this, which makes understanding the next paragraphs hard.

Environmental analysis:

The analysis of impacts of urban growth on other land-uses is clear and well presented. Some further discussion of the spatial distribution of these impacts would be very helpful. Indeed, rather than the line graphs of the time series shown in Figure 5, it might be nice to show the spatial impacts, of which land-uses would be displaced where.

The analysis of soil organic carbon feels a bit like an after-thought. Why this particular environmental metric, out of all the other possible ones? Please justify why you chose this metric. You also need to present your results in the context of the global total SOC budget, and the total greenhouse gas budget. That is, give your reader a sense of if 1.97×10^9 tons of SOC (for instance) is a lot or a little, in the global policy context around climate change.

Discussion:

This discussion section is primarily focused on the details of the urban growth modelling (Zipf's Law, etc.). This would be appropriate for a modelling paper, but for a paper in NATURE COMMUNICATIONS (particularly one that promises an analysis of environmental impacts) I would expect the focus more to be on the results of the model, and the implications for global change.

Method:

The panel analysis of urban area as a function of urban population, etc., is clearly described. At its essence, the authors estimate a regression of how per-capita urban land use at a point in time varies as a function of per-capita GDP, the level of population wide urbanization, and regional dummy variables. They then apply this regression relationship forward over time.

However, as the regression of per-capita urban land use estimated at one point in time is used to forecast changes in urban area over time, there is an important assumption here, that the per-capita urban land use at a point in time will help you predict change in urban area over time. I have two concerns here:

Statistical concern: It would seem to me more statistically valid to calculate the change in urban area (between 1990 and 2000, the two time points in the analysis), and then regress that against change in urban population, change in total GDP, etc. This is more directly related to what you want to forecast- how will a change in urban population lead to a change in urban area?

Extrapolation concern: See my Manhattan example above (or consider the example of declining former industrial cities in the U.S. and Europe). From my knowledge of historical trends, population decline does not generally lead to a substantial decline in urban area, but instead urban population density falls significantly over time. Does your training dataset (for 1990 to 2000) contain data points (regions/countries) where urban area went down over this time period? If not, it seems to me speculative to extrapolate your model beyond the domain of your training dataset? To be fair, the authors seem to be trying to account for this using Equation 6, but it isn't clear to me how it was calibrated.

Other methods comments:

Why does the projected urban area in 2010 not match the observed urban area, and why should this be corrected? Could you clarify this a bit? This is just the error of your regressions predictions?

Overall, the section "Future Land-use Simulation" is quite long and detailed. Perhaps portions of this could be incorporated in the supplementary methods section?

Figures:

Figure 1: Please define the regions used, either by expanding the text in the panels, or by saying in the legends. In particular, if there is space just write "China" instead of "CHN". And define what "LAM-L" means, as at this point in the paper the reader doesn't know (something to do with Latin America, but what does the second L stand for?). I also find the units a bit confusing. It might be more intuitive to readers if you used just km².

Figure 4: This figure is not intuitive, and is difficult to understand. Please expand the description in the Figure caption, or consider removing. For the reasons I said at the start of my remarks, I am skeptical that urban population growth will lead to urban land abandonment in the simple way modelled here.

Figure 5. It might be more intuitive to readers if you used just km².

Tables:

Table 1: Please give readers a sense of what regions these acronyms represent. If these are standard regions you have adopted from another source, then just say "Region names and acronyms follow..." in the Table legend. This table might also be more intuitive if it was just four columns, that is, with only one region per row.

Supplementary File comments:

The background on the SSPs seems a bit high-level. It would be more helpful to present, for each SSP, the couple main assumptions that are affecting urban growth. These have to do with assumptions about GDP, % urban, etc. Get a little into the assumptions of the SSPs and how they are mechanistically affecting the urban area forecasted by your model.

In general, if the presentation of the urban growth model is to be the primary finding of this paper, then much more detail is needed in this section, sufficient to allow someone to reproduce

parameterizing the model. For instance, how do the spatial variables shown in Table S2 affect the probability of urban occurrence? How were these parameters fit?

Supplementary Table 2: This table is important, but it leaves the reader with more questions than it answers. How were ecoregions used, for instance?

Reviewer #2 (Remarks to the Author):

Key results:

The paper presents global maps of urban change for a set of different SSPs. The maps have a spatial resolution of 1 km² that allows a detailed analysis of urbanization processes. Moreover losses of agricultural and natural land as well as carbon storage are assessed.

Originality and significance:

The results of the study are of great interest to scientists of many disciplines as urbanization has strong impacts on the environment but also on resource use (water, land, energy etc.). Consequently the generated maps might serve as good starting point for further impact analysis. To my knowledge these long term projections (up to 2100) in such a high spatial resolution are unique, so far. What I am missing is a comparison of the results and methodology to other global approaches to simulate urban growth (e.g. Seto et al., 2012) both in the discussion and introduction section. The authors should also revise their statement on page 3 that their land-use product is the first one that is based on the SSP framework as there are already other studies available (e.g. Popp et al., 2018).

Data & methodology:

The FLUS model was adapted to calculate urban growth processes on the global scale. Overall the modelling approach is well documented here and in other papers. Nevertheless there is a number of elements of the modelling process that need further explanation.

Regarding the calculation of land demand, the authors should explain in more detail their reason for adjusting the demand according to equations 4 and 5. Is it a kind of Bias-correction? Why does this correction refer to the previous simulation step?

Also the rationale of equation 6 is unclear to me. Why is the adjustment parameter set to 0.6? Is it a model calibration?

It would greatly enhance the quality of the paper if the authors would add some experiments regarding model sensitivity (see below).

Moreover the authors should explain in more detail their selection of spatial variables. Why are these important for the simulation of urbanization processes? Which data sets did you use? Presenting only an overview table in the supplement is insufficient when a new model is described.

Model validation is done on regional level with the Figure of Merit (FoM) method. Here the authors should explain why the FoM values are relatively low. From a calibrated data driven model I would have expected a much higher accuracy for a 15 year simulation period.

The presentation of the simulation results should be improved. I would have expected a global map for each SSP at least in the supplement. I suggest showing maps for China, USA and LAM-L in

order to become consistent with the presented data in the land demand section. An overview table of the calculated magnitudes of change for all world regions should also be added. Last but not least the full names of the regions should be added to Supplementary Figure 2.

I also don't understand the content of figure 2 in the main text. In which sense does it really depict likelihood? Is the focal statistics just an instrument for better visualization?

For figure 3 I propose to enlarge the spatial extent or to select regions where differences between the scenarios are more pronounced.

Appropriate use of statistics and treatment of uncertainties:

The authors do not address uncertainties of their results. I strongly recommend improving the paper by including at least a qualitative discussion of potential sources of uncertainties related to input data and model assumptions.

Suggested improvements:

The paper only shows results from a scenario analysis. A proper model testing e.g. with regard to uncertainties and sensitivities is missing, but in my opinion would be necessary as a new model approach is presented.

Moreover the discussion needs to be revised substantially. Here the authors should (1) reflect advantages and disadvantages of their modelling approach compared to other models (not only related to spatial resolution but also regarding process representation), (2) discuss model and data uncertainties, and most importantly (3) interpret and reflect both the calculated urbanization patterns and environmental consequences in more detail.

References: Does this manuscript reference previous literature appropriately? If not, what references should be included or excluded?

There is a whole bunch of literature regarding modelling of SSP related scenarios that should be included.

Clarity and context:

I already pointed out that both the introduction and discussion need to be revised.

My overall recommendation is that the authors should substantially revise their manuscript before publication. Most importantly they have to add further simulation experiments to explore the model behaviour in more detail.

References:

Popp, A., Calvin, K., Fujimori, S., Havlik, P., Humpenöder, F., Stehfest, E., ... & Hasegawa, T. (2017). Land-use futures in the shared socio-economic pathways. *Global Environmental Change*, 42, 331-345.

Seto, K. C., Güneralp, B., & Hutyra, L. R. (2012). Global forecasts of urban expansion to 2030 and

direct impacts on biodiversity and carbon pools. *Proceedings of the National Academy of Sciences*, 109(40), 16083-16088.

Reviewer #3 (Remarks to the Author):

This manuscript reports on an application of the Future Land-Use Simulation (FLUS) model at the global scale, using the shared socioeconomic pathways (SSPs) scenarios that are now adopted by the Coupled Model Intercomparison Project Phase 6 (CMIP6). Within the SSP scenario framework, the authors also report on related findings: urban land distribution and form, and environmental effects.

The novelty of this work lies in the global application of a land use change model at a 1km resolution that is consistent with the SSPs. This product will undoubtedly be of interest to the climate modeling community, but the authors miss a significant opportunity to learn and communicate about global urban land cover change trends and modeling challenges. As it currently stands, this manuscript reads as a technical exercise in modeling and GIS analysis, with little depth or critical interpretation. For these reasons, elaborated below, I recommend that this paper be rejected, but further work might justify a resubmission.

Central criticisms:

The authors present their modeling work largely as a technical exercise, without much critical discussion of how their modeling approach and assumptions will influence the results. Sohl et al. (2016), for example, demonstrate the low level of agreement among land use change models, even when they are informed by the same set of scenario assumptions. This indicates that the assumptions that are implicit with a particular modeling approach (or even within a particular modeling team) has a strong influence on the outcome. This is an issue that land change modelers need to consider and discuss when presenting their work. To point to a specific example in this manuscript, the authors state in the abstract that “a patch-based urban growth strategy is necessary...for the distribution of urban land-use to be consistent with Zipf’s law.” While the authors acknowledge that Zipf’s law is a primary guiding assumption, and they point to the relevant literature, they do not discuss the implications of this assumption – aside from the FoM comparisons presented in Table 1.

Similarly, there is no comprehensive discussion of error. Figure 1, for example, does not present errors bars around their estimates. Likewise, the use of FoM (Table 1) should be defined and interpreted more clearly. The authors state that FoM values “usually range from 10 percent to 30 percent” in other simulation studies, but readers should be given an indication of what the FoM would be for a “perfect fit.” This would allow readers to evaluate and contextualize the significance of overall model accuracy by region, and the significance of the contribution of the patch-based strategy.

The above comment also applies to what seems to be a key input into the model, the probability of occurrence surfaces. The input variables (listed in Supplementary Table 2) will have their own error sources – temporally, for example, they range from 2000 to 2015, so forecasts out to 2100 will compound errors related to temporal mismatch. As an aside, I do not think the authors provide enough background to reproduce this layer, noting in the methods section that the probability of occurrence surfaces are estimated “with the help of our own software.”

Another methodological choice that was made with little discussion is the choice of 32 modeling regions. While the authors may be constrained to this choice by the SSPs, it is worth discussing

whether or not these regions make sense from an urban modeling perspective.

When presenting results of the impacts of urban expansion on the environment, the authors make some very broad statements that may not be justified based on a simple analysis of land use change trajectories under the SSP scenarios. Statements such as "it is also apparent that excessive urban growth may lead to serious issues such as excessive energy consumption," "slow urban growth reflects the stagnation of the development of socio-economic factors...a particularly detrimental impact on developing countries," "taking the green road (SSP1) can maintain a city's development with enough impetus while avoiding chaos" are unfounded given the data the authors are referencing. A more nuanced discussion here is needed.

The authors frequently use imprecise or unclear language. For example, the use of the term "urban land use" is problematic, since there is no differentiation of "uses." "Urban land cover" may be a more appropriate phrase, although ultimately the authors should define what "urban" encompasses. "Urbanization" is also used in the methods section, and I infer this to mean the proportion of the population living in an urban area, but it is not defined. Other confusing or poorly defined terms and concepts include: polarized urban form, dispersive patch size distribution, dynamical urban land-use change (vs. dynamic, which is also used in similar contexts), panel data regression method.

Finally, the authors should explicitly state whether they are able to incorporate climate change-land use/cover change feedbacks. I'm guessing this is not the case, but it is worth stating and noting it as a potential limitation – and an avenue for future research.

I realize that many of the concerns I expressed regarding methods are answered in some of the background publications the authors reference. They are clearly building on years of research and development around the FLUS model. However, this manuscript currently does not stand alone, and in my opinion the authors should improve the communication of their approach and results such that an informed reader can reasonably follow along without delving into the more detailed treatments.

Reference: Sohl, TL, MC Wimberly, VC Radeloff, DM Theobald, and BM Sleeter. Divergent projections of future land use in the United States arising from different models and scenarios. *Ecological Modelling* 337(2016) 281-297. <http://dx.doi.org/10.1016/j.ecolmodel.2016.07.016>

Reviewers' comments:

Reviewer #1 (Remarks to the Author):

General thoughts:

This paper deals with an important topic (urban growth), and presents some novel model results. However, I have several major concerns:

Major concerns:

1.) The paper's title suggests a focus on the environmental impacts of urban growth, but the bulk of the paper is about the details of urban growth forecasting (e.g., Zipf's law or not, how much urban area in which region, etc.). If the goal of the paper is to present the details of urban growth forecasting, then make that the focus of the title and drop the brief mention of things like soil organic carbon. On the other hand, if this paper is trying to really focus on environmental impacts, then they need much more coverage in the paper.

Response:

We have revised the title as "Global Projections of Future Urban Land Expansion under Shared Socioeconomic Pathways" to better reflect the focus of our research.

We followed the suggestion and have removed the analysis of soil organic carbon. We now focus on the forecasts of urban expansion under the SSP scenarios and discussed the potential social and environmental impacts regarding urban shrinkage pressure, land cover changes and food production. We hope these revisions can make the paper clearer and better.

2.) This paper says that the 1km urban area forecasts were developed for this paper. This appears to be incorrect, since I believe (although I am not certain) these are the same data published in the Li et al. (2017) paper in ANNALS of THE AMERICAN ASSOCIATION OF GEOGRAPHERS. There's nothing wrong with using this dataset for this new analysis of urban growth impacts, but it should be clear (in the Abstract and elsewhere in the text) that these are the same 1km datasets that have been previously published on. For instance, in the main text it says "This study creates a global urban land-use simulation product", when it appears this was largely the same as the dataset published in the 2017 paper. The authors need to clarify if this is a new dataset, or the same as the one in the ANNALS paper.

Response:

The presented 1-km urban land forecasts are different from the dataset developed by Li et al. (2017). Our forecasts are based on the latest SSP scenarios adopted by CMIP6, while the dataset by Li et al. (2017) is under the SRES scenarios used in the IPCC AR3. The advantages of using SSPs have been explained as follows:

"In our projections, we follow the recent released scenario framework of the shared socioeconomic pathways (SSPs), which describe how global society, demographics, and economics will change in the coming century concerning policy assumptions and the socio-economic narrative (e.g., energy demand and supply and technological change)^{12, 13}. These SSPs allow us to conduct unified and comparable multi-scenario urban simulations^{12, 14, 15}...The

SSP framework is a critical component in the ongoing IPCC assessment of global climate change^{15, 21}.”

Compared to the existing dataset of global land cover under SSPs, our dataset has a finer spatial resolution. For instance, Popp et al. (2017) have estimated the amount of urban land area under the SSP narratives across different regions, but the study lacked spatial details. To the best of our knowledge, our results are the world’s first 1-km resolution maps of future global urban land under the SSP framework.

3.) The authors present an impressive forecasting methodology for urban growth, which would allow them to estimate the uncertainty in their estimates, both during the panel regression stage (which would allow confidence intervals of urban area by region) and at the pixel-level stage (since pixels are shown as having a probability of being urban, there is presumably some estimate of uncertainty from the cellular automaton modelling). However, these uncertainties are not presented in the paper. If this paper is to be suitable for NATURE COMMUNICATIONS, then uncertain intervals for quantities like urban area need to be shown. To be clear, the authors do a good job of showing how the different SSPs influence urban area. I am not talking about the variation among SSPs, I am talking about the uncertainty in estimation of urban area within each SSP.

Response:

We’ve added the uncertainty analysis for urban land demand estimation and spatial locations. Specifically, we forecast the urban land demand under the 95% confidence intervals for the panel regression model. The results are shown in Fig. 1 which has been added in the revised manuscript.

Fig. 1 | Projections of urban land demand globally and in the representative macro regions for 2010–2100 under the SSP scenarios. The region LAM-L consists of the low-income countries of Latin America, including Belize, Guatemala, Haiti, Honduras, and Nicaragua. The shaded areas represent the 95% confidence intervals of the projected urban land demand.

For the spatial simulation, multiple experiments were conducted to identify the modeling uncertainty. We run our model for 100 times and overlap the results of these simulations. We assume that areas (in grids) with higher agreements of simulated growth among the 100 runs of simulations are more certain to be urbanized in the future. For instance, a grid that has been simulated as urban in 70 out of the 100 runs is assumed more likely to become urbanized than another grid that has been simulated as urban in, for example, 50 out of the 100 runs. These results are shown in Fig. 3 and have been added in the revised manuscript.

Fig. 3 | Uncertainty in the simulation results of three major metropolitan areas in 2100. This figure shows the likelihood of each grid to become urban, which is estimated by overlapping the results of 100 simulation runs. Grids with higher agreements of simulated growth among the 100 simulation runs are assumed more likely to become urbanized.

4.) The paper lacks a clear statement of the hypothesis that it is testing or question it is answering. Please state this clearly in the introduction section of the paper.

Response:

We've clarified the research aims and contribution of our study in the revised manuscript:

"Urban land is expanding even faster than urban population², exerting a profound impact on biodiversity conservation and the water, carbon, aerosol, and nitrogen cycles in the climate system at the local and global scales³⁻⁵...Therefore, a proper understanding of how future urban land change will affect other land covers is important to alleviate the social and environmental problems that challenge the sustainable developments of human societies. Here, we present spatially explicit projections of global urban land expansion from 2015 to 2100 and discuss its impacts."

"We follow the recent released scenario framework of the shared socioeconomic pathways (SSPs), which describe how global society, demographics, and economics will change in the coming century concerning policy assumptions and the socio-economic narrative (e.g., energy demand and supply and technological change)^{12, 13}. These SSPs allow us to conduct unified and comparable multi-scenario urban simulations^{12, 14, 15}...The lack of sufficient spatial details in the existing SSP scenario projections may create uncertainties in environmental impact assessments²⁶...Our projections feature a fine spatial resolution of 1 km that preserves spatial details and avoids potential distortions in urban land patterns...To the best of our knowledge, our results are the world's first 1-km resolution maps of future global urban land under the SSP framework."

5.) The authors make a very strong assumption that urban population decline will result in urban area decline. This result seems to me to happen because they extrapolate their panel regressions (predicting per-capita urban land use at a point in time) to estimate change in urban area over time. This seems to me statistically questionable (see my comments below) and also at odds with the historical data (see my comments below about Manhattan, and about shrinking cities in the US and Europe). The authors do seem to try to mitigate this assumption with equation 6, but I am unclear how this was calibrated.

Response:

We've revised the assumption of our research, and the equation 6 in the original manuscript has been deleted. In the revised experiments, we just estimated the pressure of urban shrinkage instead of spatially explicitly predicting the shrinkage. We assume that the land conversion of non-urban land to urban land is irreversible. If the estimated urban area of a region is smaller than its current one, then the existing area will remain unchanged because its demand can be fulfilled by the existing areas.

The declining urban land demand is mainly owing to the decrease of population and slowdown of economic growth. We suspect that regions with declining urban land demand will suffer from the pressure of urban shrinkage. This is inspired by empirical studies, such as Wiechmann et al. (2012), Haase et al. (2014) and Long and Wu (2016), that have demonstrated some individual cases of urban shrinkage in the form of land abandonment. Therefore, based on the future projections of urban growth trends, we defined the pressure of urban shrinkage as the percentage of the area of "urban land surplus" over the area of existing urban land, ranging from 0 to 100%. We found that China will face the most severe pressure of urban shrinkage compared to other countries around the world. South America and almost all Asian areas will encounter substantial pressure of urban shrinkage in the SSP1 and SSP4 scenarios, while European countries and North America will face increased pressure of urban shrinkage in the SSP3 scenario. More detailed of the revised contents are as follows:

"Noticeable trends in declining urban land demand are observed in our scenario projections, which is mainly because of the decrease in population. A typical example is China (Fig. 1b), in which urban land demand sharply decreases in all scenarios after the 2040s/2050s due to the declining population (Fig. 5). We suspect that regions with declining urban land demand will suffer from the pressure of urban shrinkage. Urban shrinkage does not necessarily refer to a massive land conversion from urban to non-urban areas, although some individual cases of green recovery from urban land can be found in contemporary Germany at the block scale³⁸. Rather, in the context of population declines, urban shrinkage is more likely to manifest due to abandonments of already built-up areas. Shrinking cities due to the loss of their population have actually been occurring in some countries around world^{39,40}, even in the rapid urbanizing country of China⁴¹."

"However, to address the decline in urban land demand in the spatial simulations, we assume that the land conversion of non-urban land to urban land is irreversible and that the massive conversion of urban land to non-urban land is not allowed. For a certain region, if its estimated

urban land demand is smaller than that in an area that is already built-up, then no changes are simulated for this region, and the spatial extent of existing urban land also remains unchanged. Subsequently, for regions experiencing a decline in urban land demand, we assess their pressure on urban shrinkage, which is defined as the percentage of the area of “urban land surplus” over the area of existing urban land, ranging from 0 to 100%. The results (Fig. 6) suggest that China will no doubt face the most severe pressure of urban shrinkage compared to other countries around the world. South America and almost all Asian areas will encounter substantial pressure of urban shrinkage in the SSP1 and SSP4 scenarios, while European countries and North America will face increased pressure of urban shrinkage in the SSP3 scenario.”

6.) The manuscript needs a more clear separation in the manuscript between methods and the results, where possible. Right now, substantial discussion of modelling methods is mixed in with the text presenting the model results.

Response:

We’ve revised the structure of the manuscript accordingly. Contents of methods and the results have been clearly separated in the revised manuscript. Please see the revised main text for more details.

Other comments about the manuscript:

I don’t know the submission guidelines for this journal, but I would have preferred that there were page and line numbers. Makes the tasks of reviewers easier.

Response:

We’ve added the page and line numbers in the revised manuscript.

Throughout the paper, suggest spelling out "China" rather than "CHN", as it is only two extra characters!

Response:

We have spelled out "China" throughout the paper.

Title: The phrase "environmental effects" is rather broad, and can suggest all kinds of environmental considerations not focused on in this paper. I would suggest replacing with a phrase describing the specific environmental issues tested. As the paper is written, much of the discussion is about urban area and the area of other land uses it displaces (grassland, forest, etc.).

Response:

We have revised the title as “Global Projections of Future Urban Land Expansion under Shared Socioeconomic Pathways”. We followed the suggestion and have removed the analysis of soil organic carbon. We now focus on the forecasts of urban expansion under the SSP scenarios. We also analyzed the areas of other land uses the expanded urban land displaces and discussed the

potential impacts on food production. We hope these revisions can make the paper clearer and better.

Abstract:

Overall, I worry this abstract will not be comprehensible to the average reader of this journal, although specialists in the field of urban growth modelling will understand it fine. I would suggest you remove some of the jargon with equivalent phrases to make clear to the average reader of this journal what is scientifically novel/newsworthy about this paper. The abstract really needs to stand on its own. Some specific suggestions:

The phrase "patch-based urban growth strategy" is not clear to the reader at this point. Please use another, more intuitive phrase here. Same is true for "polarized urban form". To a lesser extent, this is true for "Zipf's Law", although most readers in this discipline of urban growth will understand the sentence.

SP1 promotes better "human-nature harmony" in what sense? This feels very imprecise for a scientific abstract.

SOC is an abbreviation- follow journal usage as to whether this needs to be spelled out the first time you use it or not.

Response: In this revised paper, the abstract has been rewritten using phrases that are more understandable to the average reader:

"Despite its small land coverage, urban land and its expansion can have profound impacts on global environments. Here, we present the scenario projections of global urban land expansion under the framework of the shared socioeconomic pathways (SSPs). Our projections feature a fine spatial resolution of 1 km that preserves spatial details and avoids potential distortions in urban land patterns. The results reveal a rapid trend in global urban expansion before the 2040s. However, China and many other Asian countries are expected to encounter substantial pressure from urban shrinkage after the 2050s. Approximately 50-63% of the newly expanded urban land is expected to comprise croplands. Global crop production will decline by approximately 1-4%, corresponding to the annual food needs of 122-1389 million people. These findings stress the importance of governing urban land development as a key measure to mitigate its negative impacts on croplands and food production."

Introductory text:

Suggest "hives of human activity" be "centers of human activity".

Suggest that "is considered a trade-off between" be rewritten as "varies between".

Response:

The Introduction section has been rewritten and the statements of "hives of human activity" and "is considered a trade-off between" have been removed.

This section needs a clear statement of the questions or hypotheses your work is testing.

We have stated more clearly our research aims and contribution of our work in this section:
“Urban land is expanding even faster than urban population², exerting a profound impact on biodiversity conservation and the water, carbon, aerosol, and nitrogen cycles in the climate system at the local and global scales³⁻⁵...Therefore, a proper understanding of how future urban land change will affect other land covers is important to alleviate the social and environmental problems that challenge the sustainable developments of human societies. Here, we present spatially explicit projections of global urban land expansion from 2015 to 2100 and discuss its impacts.”

“We follow the recent released scenario framework of the shared socioeconomic pathways (SSPs), which describe how global society, demographics, and economics will change in the coming century concerning policy assumptions and the socio-economic narrative (e.g., energy demand and supply and technological change)^{12, 13}. These SSPs allow us to conduct unified and comparable multi-scenario urban simulations^{12, 14, 15}...The lack of sufficient spatial details in the existing SSP scenario projections may create uncertainties in environmental impact assessments²⁶...Our projections feature a fine spatial resolution of 1 km that preserves spatial details and avoids potential distortions in urban land patterns...To the best of our knowledge, our results are the world’s first 1-km resolution maps of future global urban land under the SSP framework.”

Study design section:

"Adaptive inertia mechanism" and "roulette selection" are not clear to the reader at this point. If you want to use these phrases here, please define them briefly. Obviously, more detailed information can be found in the Methods section.

Response:

We have deleted these phrases in Study design section. Instead, we explained these phrases in the Supplementary Information. Please see the revised Supplementary Information for more details.

Magnitudes of urban land use section:

"takes a rocky road" is unclear and imprecise, please clarify what specifically about the SSP3 assumptions leads to lower urban growth. I know this is the shorthand name for SSP3, but a lot of readers won't, and it is more useful just to say what is happening with the assumptions of the scenario that is driving lower urban growth.

Response:

In the introduction section and Supplementary Information, we have revised the description of SSP3:

“SSP3 is a regional rivalry pathway contrary to global cooperation¹⁸.”

“SSP3 is a regional rivalry pathway contrary to global cooperation. In this scenario, as nationalism

resurges, all countries focus only on domestic or, at most, regional issues. At the same time, in this scenario, economic development suffers the most, with a 60% urbanization rate, and the investments in education and technology decline⁸.”

"However, all scenarios indicate that CHN will show a decrease in urban land use..." This result seems to me to be a result of some assumptions built in to how the panel regression of urban land use will be. Clearly, there is a correlation at a point in time between urban population and urban area, a quite predictable one you can build a regression model on. I even believe that the assumption of future urban growth being similar (e.g., at a similar urban population density) is plausible, although it is worth noting there's an assumption there when you go from historical data to forecasting. But I don't know if it plausible that a decline in urban population will lead to a decline in urban area. Are there good historical case studies of this? In Manhattan, for instance, there were more than twice as many residents in the 19th century as today, but urban area in Manhattan hasn't declined significantly. Instead, urban population density has just fallen.

Response:

We've revised the assumption about urban shrinkage. In the revised experiments, we just estimated the pressure of urban shrinkage instead of spatially explicitly predicting the shrinkage. We assume that the land conversion of non-urban land to urban land is irreversible. If the estimated urban area of a region is smaller than its current one, then the existing area will remain unchanged because its demand can be fulfilled by the existing areas.

The declining urban land demand is mainly owing to the decrease of population and slowdown of economic growth. We suspect that regions with declining urban land demand will suffer from the pressure of urban shrinkage. This is inspired by empirical studies, such as Wiechmann et al. (2012), Haase et al. (2014) and Long and Wu (2016), that have demonstrated some individual cases of urban shrinkage in the form of land abandonment. Therefore, based on the future projections of urban growth trends, we defined the pressure of urban shrinkage as the percentage of the area of "urban land surplus" over the area of existing urban land, ranging from 0 to 100%. We found that China will face the most severe pressure of urban shrinkage compared to other countries around the world. South America and almost all Asian areas will encounter substantial pressure of urban shrinkage in the SSP1 and SSP4 scenarios, while European countries and North America will face increased pressure of urban shrinkage in the SSP3 scenario. More detailed of the revised contents are as follows:

"Noticeable trends in declining urban land demand are observed in our scenario projections, which is mainly because of the decrease in population. A typical example is China (Fig. 1b), in which urban land demand sharply decreases in all scenarios after the 2040s/2050s due to the declining population (Fig. 5). We suspect that regions with declining urban land demand will suffer from the pressure of urban shrinkage. Urban shrinkage does not necessarily refer to a massive land conversion from urban to non-urban areas, although some individual cases of green recovery from urban land can be found in contemporary Germany at the block scale³⁸. Rather, in the context of population declines, urban shrinkage is more likely to manifest due to abandonments of already built-up areas. Shrinking cities due to the loss of their population have

actually been occurring in some countries around world^{39, 40}, even in the rapid urbanizing country of China⁴¹.”

“However, to address the decline in urban land demand in the spatial simulations, we assume that the land conversion of non-urban land to urban land is irreversible and that the massive conversion of urban land to non-urban land is not allowed. For a certain region, if its estimated urban land demand is smaller than that in an area that is already built-up, then no changes are simulated for this region, and the spatial extent of existing urban land also remains unchanged. Subsequently, for regions experiencing a decline in urban land demand, we assess their pressure on urban shrinkage, which is defined as the percentage of the area of “urban land surplus” over the area of existing urban land, ranging from 0 to 100%. The results (Fig. 6) suggest that China will no doubt face the most severe pressure of urban shrinkage compared to other countries around the world. South America and almost all Asian areas will encounter substantial pressure of urban shrinkage in the SSP1 and SSP4 scenarios, while European countries and North America will face increased pressure of urban shrinkage in the SSP3 scenario.”

In America, we would write "fossil-fueled" rather than "fossil-fueled", but the later is fine in British English. Whatever is standard spelling for this journal is appropriate.

Response:

We have replaced "fossil-fuelled" with "fossil-fueled".

Please define "LAM-L" before use (list the countries, or give some other definition). Is it all the Latin American countries, or just those that are low income (excluding middle income countries like Brazil, Argentina, etc.)?

Response:

“LAM-L” has been explained in the caption of Fig. 1:

“The region LAM-L consists of the low-income countries of Latin America, including Belize, Guatemala, Haiti, Honduras, and Nicaragua.”

Spatial distribution of urban land use section:

Here, and elsewhere in the results sections, there are a few sentences about methods mixed in. For instance, there’s a sentence about technical details of GeoSOS-FLUS. I would suggest having a more clear separation in the manuscript between methods and the results, where possible.

Response:

We have revised the manuscript accordingly with a clearer separation between methods and the results. This section has been rewritten with the title “Spatial distribution of future urban land expansion”. This section now focuses on the spatial results of future urban land expansion and their statistical properties.

It is a simplification of Zipf’s law to say that it "reflects the fact that larger urban patches grow faster". That may be the implication in this context, but Zipf’s "law" (really just a statement of an

empirically observed trend that has held constant over many decades) is really a statement about the city size distribution. If Zipf's law is going to be featured prominently in the results, then you should define what it means more fully in the Introduction section.

Response:

We've revised the discussion relevant to Zipf's law and moved it to the Supplementary Information due to the limited length of the main text. Please see the revised Supplementary Information for more details.

Model validation and simulation performance assessments section:

This section is very detailed. For a paper that was focused on modelling methods, this would seem appropriate. But it would be usual for a paper in NATURE COMMUNICATIONS to have this level of detail on model validation, etc., in the main text. That said, it is excellent you are testing different modelling methods using historical series of urban growth.

Response:

Thanks for your appreciation. In the revised paper, this section has been removed to Supplementary Information (Section "Model performance").

For "Figure of Merit", please give the reader a brief (1 sentence) description of how to interpret this metric. Does it range between 0 and 100%? What does it represent? Many readers without a background in remote sensing or statistics may not know this, which makes understanding the next paragraphs hard.

Response:

In the Methods section, we added an introduction of "Figure of Merit":

"The agreement between the simulated and observed urban land expansion is evaluated using the FoM (Figure of Merit) indicator⁶⁰. We use this indicator because it avoids the drawback of accuracy overestimation in conventional validation metrics (e.g., the Kappa coefficient)⁶¹. FoM is calculated as the ratio of the correct predicted change to the sum of the observed change and predicted change. The value of FoM ranges from 0 to 1, with 1 representing a perfect fit between the simulated and observed changes."

The calculation of FoM has been provided in the Supplementary Information:

"Conventional accuracy indicators, such as the overall accuracy and Kappa coefficient, are used to evaluate the simulation performances of both changed and persistently non-changed simulations. This may lead to the overestimation of accuracies when evaluating the simulation performance²¹. However, FoM avoids overestimation by only focusing on the parts of land where changes have taken place^{22, 23}. This index can be mathematically expressed as the ratio of the correct predicted change to the sum of the observed change and predicted change:

$$\text{FoM} = B / (A + B + C + D) \quad (6)$$

"where A represents the error area due to observed change predicted as persistence, B is the area of correctness due to observed change predicted as change, C denotes the area of error due to observed change predicted as change in the wrong category, and D denotes the area of error

due to observed persistence predicted as change.”

Environmental analysis:

The analysis of impacts of urban growth on other land-uses is clear and well presented. Some further discussion of the spatial distribution of these impacts would be very helpful. Indeed, rather than the line graphs of the time series shown in Figure 5, it might be nice to show the spatial impacts, of which land-uses would be displaced where.

Response:

In the revised paper, due to the limitation of article length, the areas of different land covers occupied by the expanded urban land have been shown in Supplementary Figure 8~12:

“Urban encroachments on cropland will mainly occur in China, India, sub-Saharan Africa and western Europe (Supplementary Figure 12). Forest and grassland collectively contribute to 30-44% of the newly expanded global urban land. Losses in forests due to urban land are expected to occur mainly in North America, South America, western Europe, sub-Saharan Africa and Australia (Supplementary Figure 8). Major grassland declines due to urban land expansion will occur in the USA and western Europe (Supplementary Figure 9). Global wetlands, which are one of the most biologically diverse ecosystems, will be consumed by 2-4% of the newly developed global urban land. Most wetland losses will take place in the USA and sub-Saharan Africa (Supplementary Figure 10).”

Supplementary Figure 8 | Encroached forest area caused by urban expansion in each SSP scenario by 2100.

Supplementary Figure 9 | Encroached grassland area caused by urban expansion in each SSP scenario by 2100.

Supplementary Figure 10 | Encroached wetland area caused by urban expansion in each SSP scenario by 2100.

Supplementary Figure 11 | Encroached bare land area caused by urban expansion in each SSP scenario by 2100.

Supplementary Figure 12 | Encroached cropland area caused by urban expansion in each SSP scenario by 2100.

In the revised paper, we have also added an analysis of the impact of urban expansion on crop production. The results suggest that even a trivial amount of conversion from the cropland to the urban will exert substantial impacts on crop production. Notably, the projected losses in the

production of the five key food crops (i.e. rice, wheat, maize, potato and vegetables) will reach as high as 2.86%-3.91% in SSP5, equivalent to 272-1389 million people's annual food consumption (see Method). The spatial results in SSP1-5 are provided in the supplementary information file (Supplementary Figure 14~18).

The analysis of soil organic carbon feels a bit like an after-thought. Why this particular environmental metric, out of all the other possible ones? Please justify why you chose this metric. You also need to present your results in the context of the global total SOC budget, and the total greenhouse gas budget. That is, give your reader a sense of if 1.97×10^9 tons of SOC (for instance) is a lot or a little, in the global policy context around climate change.

Response:

We have deleted the analysis of SOC to avoid the confusion. In the revised paper, we focus on the impacts of future urban expansion on other land cover types, as suggested a previous comment: "Urban encroachments on cropland will mainly occur in China, India, sub-Saharan Africa and western Europe (Supplementary Figure 12). Forest and grassland collectively contribute to 30-44% of the newly expanded global urban land. Losses in forests due to urban land are expected to occur mainly in North America, South America, western Europe, sub-Saharan Africa and Australia (Supplementary Figure 8). Major grassland declines due to urban land expansion will occur in the USA and western Europe (Supplementary Figure 9). Global wetlands, which are one of the most biologically diverse ecosystems, will be consumed by 2-4% of the newly developed global urban land. Most wetland losses will take place in the USA and sub-Saharan Africa (Supplementary Figure 10)."

We further explored the impacts of urban expansion on food production:

"The results (Fig. 7) indicate that the scenarios SSP3 and SSP4 yield the smallest amount of estimated crop production losses, mainly because of the decrease in socio-economic developments and urban expansion processes. In contrast, scenario SSP5 reports the greatest amount of crop production losses due to the assumed rapid urban expansion. Overall, by 2100, global urban land expansion will cause declines of 2-3% in rice production, 1-3% in wheat production, 1-4% in maize production, 1-3% in potato production and 2-4% in vegetable production (see Supplementary Figures 14-18 for more spatial details). These losses are expected to affect the annual food needs of approximately 122-1389 million people, as estimated according to different food structures (Table 2)."

"Our projections (Fig. 7) demonstrate that future urban land expansion will cause up to 1% of loss in the global cropland area. However, the direct losses of global crop production are likely to exceed 1% and can even reach as high as 4%, depending on the specific crop types. These results suggest the importance of conserving high-quality croplands to reduce the impacts of urban expansion on food production capacity."

Discussion:

This discussion section is primarily focused on the details of the urban growth modelling (Zipf's Law, etc.). This would be appropriate for a modelling paper, but for a paper in NATURE

COMMUNICATIONS (particularly one that promises an analysis of environmental impacts) I would expect the focus more to be on the results of the model, and the implications for global change.

Response:

We have revised this section as follows:

“Based on the urban land projections, we have identified the pressure caused by future urban shrinkage for different regions across the world...These results have important implications for urban planning and management. In rapidly urbanizing countries, such as China, urban planning is usually prepared with the expectation of growth, with little or no awareness of the ongoing/upcoming challenges caused by urban shrinkage⁴¹. A direct consequence of urban shrinkage is the difficulty of sustaining the maintenance of infrastructure, which was originally built to serve a larger population, leading to declines in livelihoods and quality of life in cities. Abandonments and social disorder are also potential outcomes raised by urban shrinkage that threaten public health and safety in cities^{46, 47}. Moreover, from an economic point of view, urban shrinkage may cause economic recession because of the negative impacts on employment, industrial production and property value⁴⁸⁻⁵⁰.”

“Our projections demonstrate that most future urban land expansion occurs over croplands (50-63%) and forests (30-44%). The losses in croplands and forests will have profound impacts on ecosystem services, such as carbon sequestration, habitat provision and food supply to human societies⁵¹⁻⁵⁴. Our results show that the crop production losses induced by urban expansion will affect the annual food needs of approximately 122-1389 million people. More importantly, the crop production losses (approximately 1-4%) are disproportional to the cropland land losses (approximately 1%). Previous studies have revealed that a 2 °C global temperature increase will cause 8-14% of global maize production losses⁵⁵. In comparison, our projections suggest that urban land expansion alone will lead to 1-3% of global maize production loss, indicating the profound impacts of urban expansion on global crop production. These results imply the critical importance of conserving fertile and productive cropland. Governance and proper planning of urban land development locations and procedures is a key measure to reduce future losses in crop production^{43, 56}. However, the governance qualities appear to be moderate to low due to the fragmented governance structures in many countries that will suffer from large crop production losses⁵⁷. The effective protection of croplands, therefore, calls for increasing efforts in educating policy makers and the development of more comprehensive governance regimes⁵⁸.”

Method:

The panel analysis of urban area as a function of urban population, etc., is clearly described. At its essence, the authors estimate a regression of how per-capita urban land use at a point in time varies as a function of per-capita GDP, the level of population wide urbanization, and regional dummy variables. They then apply this regression relationship forward over time.

However, as the regression of per-capita urban land use estimated at one point in time is used to forecast changes in urban area over time, there is an important assumption here, that the per-capita urban land use at a point in time will help you predict change in urban area over time.

I have two concerns here:

Statistical concern: It would seem to me more statistically valid to calculate the change in urban area (between 1990 and 2000, the two time points in the analysis), and then regress that against change in urban population, change in total GDP, etc. This is more directly related to what you want to forecast- how will a change in urban population lead to a change in urban area?

Response:

We apologize for a mistake in the description of data in the original manuscript. We actually used four sets of data for the years 1975, 1990, 2000 and 2014 rather than only two years (i.e. 1990 and 2000). The results of the panel data regression model established using these data have passed a t-test at a confidence level of 0.01. We also illustrated the uncertainty of the results based on their 95% confidence intervals, as shown in the revised Fig. 1.

Fig. 2 | Projections of urban land demand globally and in the representative macro regions for 2010–2100 under the SSP scenarios. The region LAM-L consists of the low-income countries of Latin America, including Belize, Guatemala, Haiti, Honduras, and Nicaragua. The shaded areas represent the 95% confidence intervals of the projected urban land demand.

Extrapolation concern: See my Manhattan example above (or consider the example of declining former industrial cities in the U.S. and Europe). From my knowledge of historical trends, population decline does not generally lead to a substantial decline in urban area, but instead

urban population density falls significantly over time. Does your training dataset (for 1990 to 2000) contain data points (regions/countries) where urban area went down over this time period? If not, it seems to me speculative to extrapolate your model beyond the domain of your training dataset? To be fair, the authors seem to be trying to account for this using Equation 6, but it isn't clear to me how it was calibrated.

Response:

We've revised the assumption of our research, and the original equation 6 has been deleted. More details have been provided in a previous response to the comment with the Manhattan example.

Other methods comments:

Why does the projected urban area in 2010 not match the observed urban area, and why should this be corrected? Could you clarify this a bit? This is just the error of your regressions predictions?

Response:

In the revised manuscript, we've revised the expressions in this part and explained the implementation of these adjustments:

"We use the socioeconomic data at year 2010, which is the starting year of the SSP database, to evaluate the performance of the model for urban land demand estimation. The results yield an overall error of 10.95% at the global level. To further reduce the impacts of this error on urban simulation, we implement the following adjustments:

$$\Delta A'_{r,2010 \rightarrow t} = \frac{A'_{r,t}}{A'_{r,2010}} \quad (3)$$

$$A_{r,t} = A_{r,2010} \times \Delta A'_{r,2010 \rightarrow t} \quad (4)$$

Where $\Delta A'_{r,2010 \rightarrow t}$ is the estimated growth rates in region r from year 2010 to future year t ; $A'_{r,2010}$ and $A'_{r,t}$ are the estimated urban land demand in region r at year 2010 and future year t ; $A_{r,t}$ is the adjusted estimation of urban land demand in region r at future year t ; $A_{r,2010}$ is the observed area of urban land cover at year 2010, which is derived from the actual global urban land map at year 2010. Therefore, the original estimations of urban land demand for future years are consistently adjusted using the reference of the actual area of urban land cover at year 2010. The adjusted estimation of urban land demand $A_{r,t}$ is then used to constrain the spatial simulation of urban land expansion."

These sentences have been moved to the Supplementary Information.

Overall, the section "Future Land-use Simulation" is quite long and detailed. Perhaps portions of this could be incorporated in the supplementary methods section?

Response:

The section "Future Land-use Simulation" has been moved to Supplementary Information.

Figures:

Figure 1: Please define the regions used, either by expanding the text in the panels, or by saying in the legends. In particular, if there is space just write "China" instead of "CHN". And define what "LAM-L" means, as at this point in the paper the reader doesn't know (something to do with Latin America, but what does the second L stand for?). I also find the units a bit confusing. It might be more intuitive to readers if you used just km².

Response:

We have changed "CHN" to "China". "LAM-L" has been defined in the caption of Fig. 1: "The region LAM-L consists of the low-income countries of Latin America, including Belize, Guatemala, Haiti, Honduras, and Nicaragua." Since the global urban demands are very large (Fig. 1(a)), we use the expression "10³ km²" to avoid excessive numbers.

Figure 4: This figure is not intuitive, and is difficult to understand. Please expand the description in the Figure caption, or consider removing. For the reasons I said at the start of my remarks, I am skeptical that urban population growth will lead to urban land abandonment in the simple way modelled here.

Response:

We've revised the assumption of our research, and the original Figure 4 has been deleted. The details of the revised results of urban shrinkage have been provided in the previous responses. We suspect that regions with declining urban land demand (caused by declining population and the slowdown of economic growth) will suffer from the pressure of urban shrinkage. This is inspired by previous empirical studies such as Wiechmann et al. (2012), Haase et al. (2014) and Long and Wu (2016), which have reported contemporary cases of urban shrinkage in a form of land abandonment.

Figure 5. It might be more intuitive to readers if you used just km².

Response:

The original Fig. 5 has been deleted.

Tables:

Table 1: Please give readers a sense of what regions these acronyms represent. If these are standard regions you have adopted from another source, then just say "Region names and acronyms follow..." in the Table legend. This table might also be more intuitive if it was just four columns, that is, with only one region per row.

Response:

The original Table 1 has been revised accordingly, and moved to Supplementary Information due to the limited length of main text. We've specified the sources of the full region names at the footnote of this table: "Note: Region names and acronyms follow the official SSP dataset (<https://tntcat.iiasa.ac.at/SspDb>)."

Supplementary File comments:

The background on the SSPs seems a bit high-level. It would be more helpful to present, for each SSP, the couple main assumptions that are affecting urban growth. These have to do with assumptions about GDP, % urban, etc. Get a little into the assumptions of the SSPs and how they are mechanistically affecting the urban area forecasted by your model.

Response:

We give a more detailed introduction to the assumptions of SSPs with respect to population, GDP, and urbanization rate:

“The five SSP scenarios are located at different sites in the ‘challenge space’ and are used to represent different socio-economic development pathways⁵. SSP1 is a people-oriented sustainable pathway that uses green roads⁶. In SSP1, by the end of this century, the global urbanization rate will reach 92%, with a rapidly growing GDP. SSP2 is a middle pathway between SSP1 and SSP3⁷, and the urbanization rate and GDP values are also between the values corresponding to these two scenarios. SSP3 is a regional rivalry pathway contrary to global cooperation. In this scenario, as nationalism resurges, all countries focus only on domestic or, at most, regional issues. At the same time, in this scenario, economic development suffers the most, with a 60% urbanization rate, and the investments in education and technology decline⁸. SSP4 is a divided pathway in which inequality and stratification increase both across and within countries. However, high-tech economic sectors are highly developed⁹; thus, in this scenario, economic development is relatively better than that in SSP3, and the global urbanization rate remains the same as that in SSP1. SSP5 is a fossil-fueled development pathway, where people exploit abundant fossil fuel resources, the global economy grows at the highest speed, and the global urbanization rate reaches 92%. However, at the same time, people face severe mitigation challenges¹⁰. Notably, these only apply to the global situation. For the countries with different developments (such as developed countries, developing countries and low-income countries), their socio-economic assumptions in the same scenario are different or even the opposite. For example, developed countries have the fastest population growth in SSP5 and the slowest growth in SSP3, but low-income countries experience the opposite situation.”

In general, if the presentation of the urban growth model is to be the primary finding of this paper, then much more detail is needed in this section, sufficient to allow someone to reproduce parameterizing the model. For instance, how do the spatial variables shown in Table S2 affect the probability of urban occurrence? How were these parameters fit?

Response:

We’ve clarified the use of the spatial variables in the revised Supplementary Information:

“The estimation of the ‘probability-of-occurrence’ is based on the ANN method with a set of spatial variables. These spatial variables are selected mainly according to the findings of previous studies on urban land expansion simulation¹¹⁻¹³. They include population, GDP, distance to the city centre, distance to the road network, distance to the airport, elevation, slope and the water resource condition, which collectively cover the major socioeconomic and natural factors affecting urban land expansion¹¹⁻¹³. To take into account the ecological characteristics, we also acquire the ecoregion types from a world ecoregions map (Supplementary Table 2) and use them as nominal features in the ANN training. The corresponding data sources of all these spatial

variables are provided in Supplementary Table 2. All these variables are processed to have a spatial resolution of 1 km with a spatial reference of *WGS_1984_Cylindrical_Equal_Area*.

The estimation of the ‘probability-of-occurrence’ is carried out separately for each of the 32 macro regions. Specifically, for each macro region, a sample is collected using a stratified random sampling approach with an equal proportion of 30% for urban and non-urban grids, respectively. This sample is used as a training dataset to train an ANN that yields the corresponding classification probabilities for ‘urban’ or ‘non-urban’. The training procedure is based on the traditional back-propagation (BP) approach¹⁴. After the training procedure, for a non-urban grid i of land cover type k , its value of ‘probability-of-occurrence’ $P_{i,k}$ is represented using the ‘urban’ classification probability obtained from a trained ANN.”

Supplementary Table 2: This table is important, but it leaves the reader with more questions than it answers. How were ecoregions used, for instance?

Response:

We’ve clarified the use of the ecoregions:

“To take into account the ecological characteristics, we also acquire the ecoregion types from a world ecoregions map (Supplementary Table 2) and use them as nominal features in the ANN training.”

Reviewer #2 (Remarks to the Author):

Key results:

The paper presents global maps of urban change for a set of different SSPs. The maps have a spatial resolution of 1 km² that allows a detailed analysis of urbanization processes. Moreover losses of agricultural and natural land as well as carbon storage are assessed.

Originality and significance:

The results of the study are of great interest to scientists of many disciplines as urbanization has strong impacts on the environment but also on resource use (water, land, energy etc.). Consequently the generated maps might serve as good starting point for further impact analysis. To my knowledge these long term projections (up to 2100) in such a high spatial resolution are unique, so far. What I am missing is a comparison of the results and methodology to other global approaches to simulate urban growth (e.g. Seto et al., 2012) both in the discussion and introduction section. The authors should also revise their statement on page 3 that their land-use product is the first one that is based on the SSP framework as there are already other studies available (e.g. Popp et al., 2018).

Response:

The differences between our products and others are mainly in the scenario settings and spatial resolutions. We've briefly compared our research with others. The dataset generated by Seto et al. (2012) was based on a single scenario with a storyline that is different from the SSPs. Popp et al. (2017) predicted the magnitude of future land covers under different SSP scenarios, but these results were at the region level instead of grid level. Our results, however, are based on the SSPs and have a much finer resolution (1 km). We've clarified these differences in the revised manuscript:

"For instance, a 2030 global urban land map has been developed based on global population and economy predictions made by the United Nations⁸. The universal climate scenarios developed by the Intergovernmental Panel on Climate Change (IPCC) have also been used to simulate future changes in global and regional land covers, including an urban land category⁹⁻¹¹. In our projections, we follow the recent released scenario framework of the shared socioeconomic pathways (SSPs), which describe how global society, demographics, and economics will change in the coming century concerning policy assumptions and the socio-economic narrative (e.g., energy demand and supply and technological change)^{12, 13}. These SSPs allow us to conduct unified and comparable multi-scenario urban simulations^{12, 14, 15}."

"Although some recent studies have predicted urban land distribution under SSPs at regional scales²², relevant results at a global scale are still scarce. Popp et al.²³ estimated the amount of urban land area under the SSP narratives across different regions, but the study lacked spatial details. Other global spatial projections typically have coarse spatial resolutions between 5 arc minutes and 0.5 degrees^{24, 25}. In the latest CMIP6 (Coupled Model Intercomparison Project Phase 6) Land Use Harmonization dataset, urban land is represented fractionally, with a resolution of only 0.25 degrees...To the best of our knowledge, our results are the world's first 1-km resolution maps of future global urban land under the SSP framework."

"The technical implementation of our projections is based on the Future Land-Use Simulation (FLUS) model²⁷. This model utilizes a machine learning approach to capture the complex relationships between urban land expansion and its driving factors. This model also adopts the mechanisms of cellular automata (CA)²⁷⁻²⁹, which are capable of reflecting the complexities of path-dependence and positive feedback in the actual processes of urban land expansion³⁰."

Data & methodology:

The FLUS model was adapted to calculate urban growth processes on the global scale. Overall the modelling approach is well documented here and in other papers. Nevertheless there is a number of elements of the modelling process that need further explanation.

Regarding the calculation of land demand, the authors should explain in more detail their reason for adjusting the demand according to equations 4 and 5. Is it a kind of Bias-correction? Why does this correction refer to the previous simulation step?

Response:

In the revised manuscript, we've revised the expressions in this part and explained the implementation of these adjustments:

"We use the socioeconomic data at year 2010, which is the starting year of the SSP database, to evaluate the performance of the model for urban land demand estimation. The results yield an overall error of 10.95% at the global level. To further reduce the impacts of this error on urban simulation, we implement the following adjustments:

$$\Delta A'_{r,2010 \rightarrow t} = \frac{A'_{r,t}}{A'_{r,2010}} \quad (3)$$

$$A_{r,t} = A_{r,2010} \times \Delta A'_{r,2010 \rightarrow t} \quad (4)$$

Where $\Delta A'_{r,2010 \rightarrow t}$ is the estimated growth rates in region r from year 2010 to future year t; $A'_{r,2010}$ and $A'_{r,t}$ are the estimated urban land demand in region r at year 2010 and future year t; $A_{r,t}$ is the adjusted estimation of urban land demand in region r at future year t; $A_{r,2010}$ is the observed area of urban land cover at year 2010, which is derived from the actual global urban land map at year 2010. Therefore, the original estimations of urban land demand for future years are consistently adjusted using the reference of the actual area of urban land cover at year 2010. The adjusted estimation of urban land demand $A_{r,t}$ is then used to constrain the spatial simulation of urban land expansion."

These sentences have been moved to the Supplementary Information.

Also the rationale of equation 6 is unclear to me. Why is the adjustment parameter set to 0.6? Is it a model calibration?

Response:

We've revised the assumption of our research, and the original equation 6 has been deleted.

In the revised experiments, we just estimated the pressure of urban shrinkage instead of spatially explicitly predicting the shrinkage. We assume that the land conversion of non-urban land to urban land is irreversible. If the estimated urban area of a region is smaller than its current one, then the existing area will remain unchanged because its demand can be fulfilled by the existing areas.

The declining urban land demand is mainly owing to the decrease of population and slowdown of economic growth. We suspect that regions with declining urban land demand will suffer from the pressure of urban shrinkage. This is inspired by empirical studies, such as Wiechmann et al. (2012), Haase et al. (2014) and Long and Wu (2016), that have demonstrated some individual cases of urban shrinkage in the form of land abandonment. Therefore, based on the future projections of urban growth trends, we defined the pressure of urban shrinkage as the percentage of the area of "urban land surplus" over the area of existing urban land, ranging from 0 to 100%. We found that China will face the most severe pressure of urban shrinkage compared to other countries around the world. South America and almost all Asian areas will encounter substantial pressure of urban shrinkage in the SSP1 and SSP4 scenarios, while European countries and North America will face increased pressure of urban shrinkage in the SSP3 scenario.

It would greatly enhance the quality of the paper if the authors would add some experiments regarding model sensitivity (see below).

Moreover the authors should explain in more detail their selection of spatial variables. Why are these important for the simulation of urbanization processes? Which data sets did you use? Presenting only an overview table in the supplement is insufficient when a new model is described.

Response:

We explained the choice of spatial variables, which is mainly based on the findings of land cover change in the literature. We've also provide the data sources of these variables:

"The estimation of the 'probability-of-occurrence' is based on the ANN method with a set of spatial variables. These spatial variables are selected mainly according to the findings of previous studies on urban land expansion simulation¹¹⁻¹³. They include population, GDP, distance to the city centre, distance to the road network, distance to the airport, elevation, slope and the water resource condition, which collectively cover the major socioeconomic and natural factors affecting urban land expansion¹¹⁻¹³. To take into account the ecological characteristics, we also acquire the ecoregion types from a world ecoregions map (Supplementary Table 2) and use them as nominal features in the ANN training. The corresponding data sources of all these spatial variables are provided in Supplementary Table 2. All these variables are processed to have a spatial resolution of 1 km with a spatial reference of *WGS_1984_Cylindrical_Equal_Area*."

Model validation is done on regional level with the Figure of Merit (FoM) method. Here the authors should explain why the FoM values are relatively low. From a calibrated data driven model I would have expected a much higher accuracy for a 15 year simulation period.

Response:

We've explained the reason in Supplementary Information:

"Conventional accuracy indicators, such as the overall accuracy and Kappa coefficient, are used to evaluate the simulation performances of both changed and persistently non-changed simulations. This may lead to the overestimation of accuracies when evaluating the simulation performance²⁴. However, FoM avoids overestimation by only focusing on the parts of land where changes have taken place^{25,26}."

"We use FoM, which is an indicator ranging from 0% to 100% that reflects the simulation accuracy by focusing only on the part of the land that has changed, to measure the performance of the simulation results for urban land cover change from 2000 to 2015^{25,26}. Supplementary Table 1 shows the FoM values of the two methods for simulating global urban land cover change in 32 regions from 2000 to 2015. These methods, with and without the patch-based strategy, yield mean FoM values of 24.48% and 24.42% for the 32 regions, respectively. Our results are similar to those of other case studies on land cover change modelling. Previous comparative analysis has demonstrated that the common range of FoM values is between 10% and 30% for existing land cover change models^{11,22}. A recent global land cover simulation conducted by Li et al.¹¹ also reported similar results of FoM values, which ranged from 10% to 29%, with a mean value of 19%. Additionally, empirical studies have found that the value of FoM is strongly affected by the relative amount of observed net change²⁵. Small FoM values were usually found in cases

with a small amount of observed net change. For instance, Pontius et al.²⁵ revealed that the FoM value may decrease to less than 8% if the observed net change accounts for less than 2% of the total land area. In our case, however, the observed urban land growth only accounts for 0.06% of the global terrestrial area, while the FoM values of our model exceed 24%. In this sense, the performance of our model is reliable, as indicated by the relatively higher FoM values, given a small amount of observed net change.”

The presentation of the simulation results should be improved. I would have expected a global map for each SSP at least in the supplement. I suggest showing maps for China, USA and LAM-L in order to become consistent with the presented data in the land demand section. An overview table of the calculated magnitudes of change for all world regions should also be added. Last but not least the full names of the regions should be added to Supplementary Figure 2.

Response: We have added the detailed simulation results for each SSP scenario into the appendix (Supplementary Figure 3~7). We followed the suggestion and show maps for China, the USA and LAM-L to maintain the consistency with the presented data in the land demand section.

Supplementary Figure 3 | Simulation results in 2100 in SSP1. a) eastern China; b) eastern USA; c) LAM-L. Focal statistics are used as an instrument for better visualization.

Supplementary Figure 4 | Simulation results in 2100 in SSP2. a) eastern China; b) eastern USA; c) LAM-L. Focal statistics are used as an instrument for better visualization.

Supplementary Figure 5 | Simulation results in 2100 in SSP3. a) eastern China; b) eastern USA; c) LAM-L.

Focal statistics are used as an instrument for better visualization.

Supplementary Figure 6 | Simulation results in 2100 in SSP4. a) eastern China; b) eastern USA; c) LAM-L. Focal statistics are used as an instrument for better visualization.

Supplementary Figure 7 | Simulation results in 2100 in SSP5. a) eastern China; b) eastern USA; c) LAM-L. Focal statistics are used as an instrument for better visualization.

An overview table is added into Supplementary Information, to display urban growth of each region from 2015 to 2100 (Supplementary Table 3).

Supplementary Table 3 | Simulated urban growth between 2015 and 2100 in each SSP

Region	Urban growth between 2015 and 2100 (km ²) / Growth rate since 2015 (%)				
	SSP1	SSP2	SSP3	SSP4	SSP5
ANUZ	14039(101.8)	13876(100.6)	3317(24.0)	9874(71.6)	32503(235.6)
BRA	4647(18.6)	6794(27.3)	14124(56.7)	3509(14.1)	5039(20.2)
CAN	10212(94)	9631(88.6)	1438(13.2)	5949(54.7)	26398(242.9)
CAS	4733(44.3)	5470(51.2)	10535(98.6)	4326(40.5)	4102(38.4)
CHN	33834(32.6)	28359(27.3)	21352(20.5)	27689(26.6)	38289(36.8)
EEU	882(28.5)	767(24.8)	909(29.3)	616(19.9)	1243(40.1)
EEU-FSU	938(3.8)	655(2.7)	2824(11.5)	414(1.7)	1831(7.5)
EFTA	6980(151.2)	6415(138.9)	1349(29.2)	4161(90.1)	15847(343.2)
EU12-H	3985(27.3)	2146(14.7)	343(2.4)	1296(8.9)	13535(92.8)
EU12-M	1108(11.3)	679(7.0)	1031(10.5)	726(7.4)	1254(12.8)
EU15	56291(65.6)	50391(58.7)	2970(3.5)	24699(28.8)	145923(170)

IDN	5679(36.5)	5997(38.6)	5921(38.1)	4233(27.2)	5939(38.2)
IND	28693(123.9)	33531(144.8)	39573(170.8)	25246(109)	33195(143.3)
JPN	775(3.2)	125(0.5)	0(0.0)	81(0.3)	5660(23.4)
KOR	326(9.9)	220(6.7)	102(3.1)	152(4.6)	1140(34.6)
LAM-L	1719(88.3)	2515(129.2)	4380(225.1)	5090(261.6)	1403(72.1)
LAM-M	8079(33.4)	12027(49.7)	25469(105.3)	7513(31.1)	7922(32.8)
MEA-H	8601(101.8)	12399(146.8)	15292(181.1)	15179(179.7)	13878(164.3)
MEA-M	10646(100.7)	15642(148)	24462(231.4)	26334(249.1)	12203(115.4)
MEX	4449(35.7)	7667(61.6)	18850(151.4)	3915(31.4)	3743(30.1)
NAF	4916(57.6)	6341(74.3)	10416(122.1)	4235(49.6)	4837(56.7)
OAS-CPA	2084(54.4)	1967(51.3)	2493(65.0)	1855(48.4)	2026(52.9)
OAS-L	4557(81.2)	5107(91.0)	7630(136.0)	7936(141.5)	4445(79.2)
OAS-M	4187(50.9)	4726(57.4)	6320(76.8)	3307(40.2)	5097(61.9)
PAK	5988(151.7)	8874(224.8)	11507(291.5)	17638(446.8)	6116(154.9)
RUS	2317(7.6)	4557(15.0)	8235(27.1)	1514(5.0)	5652(18.6)
SAF	3159(43.8)	3238(44.9)	3776(52.4)	1664(23.1)	4233(58.7)
SSA-L	61762(262.1)	76787(325.9)	87343(370.7)	135336(574.4)	67784(287.7)
SSA-M	1647(75.8)	2102(96.8)	2754(126.8)	3226(148.5)	1741(80.1)
TUR	3583(49.6)	5416(74.9)	8350(115.5)	3173(43.9)	4567(63.2)
USA	106038(76.6)	97342(70.3)	17113(12.4)	54177(39.1)	262734(189.8)
global	406853(60.8)	431765(64.5)	360177(53.8)	405064(60.5)	740279(110.6)

Note: The 32 regions are defined following the official SSP dataset (<https://tntcat.iiasa.ac.at/SspDb>).

We have added a description of the definitions of the regions in the caption of Supplementary Figure 2:

“The 32 regions are defined following the official SSP dataset (<https://tntcat.iiasa.ac.at/SspDb>).”

I also don’t understand the content of figure 2 in the main text. In which sense does it really depict likelihood? Is the focal statistics just an instrument for better visualization?

Response: We’ve revised the caption for this figure. Figure 2 is the results of focal statistics of urban land. We used focal statistics in order to provide a better visualization for our results.

Fig. 3 | The simulated urban land maps in 2100 for (a) China, (b) the USA and (c) LAM-L in scenarios SSP3 and SSP5. Focal statistics have been applied to the simulated maps for better visualization. The results show that China and the USA experience more urban land expansion in scenario SSP5 than in SSP3. In contrast, the LAM-L region experiences greater urban land expansion in scenario SSP3 than in SSP5.

For figure 3 I propose to enlarge the spatial extent or to select regions where differences between the scenarios are more pronounced.

Response: The original Figure 3 has been deleted. We enlarged the spatial extent to show the simulation results in Supplementary Figures 3~7.

Appropriate use of statistics and treatment of uncertainties:

The authors do not address uncertainties of their results. I strongly recommend improving the paper by including at least a qualitative discussion of potential sources of uncertainties related to input data and model assumptions.

Response:

We've added the uncertainty analysis for urban land demand estimation and spatial locations. Specifically, we forecast the urban land demand under the 95% confidence intervals for the panel regression model. The results are shown in Fig. 1 which has been added in the revised manuscript.

Fig. 4 | Projections of urban land demand globally and in the representative macro regions for 2010–2100 under the SSP scenarios. The region LAM-L consists of the low-income countries of Latin America, including Belize, Guatemala, Haiti, Honduras, and Nicaragua. The shaded areas represent the 95% confidence intervals of the projected urban land demand.

For the spatial simulation, multiple experiments were conducted to identify the modeling uncertainty. We run our model for 100 times and overlap the results of these simulations. We assume that areas (in grids) with higher agreements of simulated growth among the 100 runs of simulations are more certain to be urbanized in the future. For instance, a grid that has been simulated as urban in 70 out of the 100 runs is assumed more likely to become urbanized than another grid that has been simulated as urban in, for example, 50 out of the 100 runs. These results are shown in Fig. 3 and have been added in the revised manuscript.

Fig. 3 | Uncertainty in the simulation results of three major metropolitan areas in 2100. This figure shows the likelihood of each grid to become urban, which is estimated by overlapping the results of 100 simulation runs. Grids with higher agreements of simulated growth among the 100 simulation runs are assumed more likely to become urbanized.

Moreover, we discuss potential sources of uncertainties related to input data and model assumptions:

“This model adopts several assumptions: (1) Urban conversion on other land cover types is irreversible and hence any recovery from urban land to non-urban land is not allowed; (2) For any regions, if the estimated urban land demand is smaller than the existing area of land that is already built-up, then no changes are simulated; (3) The spatial driving factors are held constant in future projections of urban land expansion.”

“There are several limitations in our projections. First, we do not address the potential errors in future urban land expansion that are caused by the misclassifications in historical urban land maps, which were acquired from the GHSL dataset. Second, we hold the spatial driving factors of urban land expansion constant in the scenario projections, which is mainly because of the difficulty in the reliable prediction of, for example, changes in urban infrastructure. Third, we do not consider the impacts of future climate change on land-use change. This limitation can be alleviated in our future work by applying the projections under the integrated scenarios of RCP-SSP, which take into account the feedbacks of global climate changes.”

Suggested improvements:

The paper only shows results from a scenario analysis. A proper model testing e.g. with regard to uncertainties and sensitivities is missing, but in my opinion would be necessary as a new model approach is presented.

Response:

We've added the uncertainty analysis for urban land demand estimation and spatial locations. Please see the detailed response to the previous comments above.

Moreover the discussion needs to be revised substantially. Here the authors should (1) reflect advantages and disadvantages of their modelling approach compared to other models (not only related to spatial resolution but also regarding process representation), (2) discuss model and data uncertainties, and most importantly (3) interpret and reflect both the calculated urbanization patterns and environmental consequences in more detail.

Response:

(1) We discuss the differences between our projections and others:

“Projections of urban land patterns require established scenarios that represent possible future socio-economic and environmental conditions. For instance, a 2030 global urban land map has been developed based on global population and economy predictions made by the United Nations⁸. The universal climate scenarios developed by the Intergovernmental Panel on Climate Change (IPCC) have also been used to simulate future changes in global and regional land covers, including an urban land category⁹⁻¹¹. In our projections, we follow the recent released scenario framework of the shared socioeconomic pathways (SSPs), which describe how global society, demographics, and economics will change in the coming century concerning policy assumptions and the socio-economic narrative (e.g., energy demand and supply and technological change)^{12,13}. These SSPs allow us to conduct unified and comparable multi-scenario urban simulations^{12,14,15}.”

“Although some recent studies have predicted urban land distribution under SSPs at regional scales²², relevant results at a global scale are still scarce. Popp et al.²³ estimated the amount of urban land area under the SSP narratives across different regions, but the study lacked spatial details. Other global spatial projections typically have coarse spatial resolutions between 5 arc minutes and 0.5 degrees^{24,25}. In the latest CMIP6 (Coupled Model Intercomparison Project Phase 6) Land Use Harmonization dataset, urban land is represented fractionally, with a resolution of only 0.25 degrees. The lack of sufficient spatial details in the existing SSP scenario projections may create uncertainties in environmental impact assessments²⁶. A recent study has warned that the common land cover product could produce major distortions in land cover patterns at a 10-km resolution⁹. Our projections have a fine spatial resolution of 1 km, which preserves spatial details and can avoid the distortions in global urban land patterns. To the best of our knowledge, our results are the world's first 1-km resolution maps of future global urban land under the SSP framework.”

(2) We added the discussion on uncertainties:

“There are several limitations in our projections. First, we do not address the potential errors in future urban land expansion that are caused by the misclassifications in historical urban land maps, which were acquired from the GHSL dataset. Second, we hold the spatial driving factors of urban land expansion constant in the scenario projections, which is mainly because of the difficulty in the reliable prediction of, for example, changes in urban infrastructure. Third, we do not consider the impacts of future climate change on land-use change. This limitation can be alleviated by applying the projections under the integrated scenarios of RCP-SSP, which take into

account the feedbacks of global climate changes.”

(3) We have enhanced the scenario analysis of future urban land expansion and its impacts: “Based on the urban land projections, we have identified the pressure caused by future urban shrinkage for different regions across the world. Urban shrinkage occurrences largely correspond to low birth rates and population ageing. Typical regions that will suffer from substantial pressure from urban shrinkage mainly include China and many other Asian countries. North America, South America and western Europe experience great pressure from urban shrinkage mainly in scenarios SSP3 and SSP4. These results have important implications for urban planning and management. In rapidly urbanizing countries, such as China, urban planning is usually prepared with the expectation of growth, with little or no awareness of the ongoing/upcoming challenges caused by urban shrinkage⁴¹. A direct consequence of urban shrinkage is the difficulty of sustaining the maintenance of infrastructure, which was originally built to serve a larger population, leading to declines in livelihoods and quality of life in cities. Abandonments and social disorder are also potential outcomes raised by urban shrinkage that threaten public health and safety in cities^{46, 47}. Moreover, from an economic point of view, urban shrinkage may cause economic recession because of the negative impacts on employment, industrial production and property value⁴⁸⁻⁵⁰.”

“Despite the potential urban shrinkage in some regions of the world, global urban land is expected to experience rapid growth and modify the global landscape before the 2040s. Our projections demonstrate that most future urban land expansion occurs over croplands (50-63%) and forests (30-44%). The losses in croplands and forests will have profound impacts on ecosystem services, such as carbon sequestration, habitat provision and food supply to human societies⁵¹⁻⁵⁴. Our results show that the crop production losses induced by urban expansion will affect the annual food needs of approximately 122-1389 million people. More importantly, the crop production losses (approximately 1-4%) are disproportional to the cropland land losses (approximately 1%). Previous studies have revealed that a 2 °C global temperature increase will cause 8-14% of global maize production losses⁵⁵. In comparison, our projections suggest that urban land expansion alone will lead to 1-3% of global maize production loss, indicating the profound impacts of urban expansion on global crop production. These results imply the critical importance of conserving fertile and productive cropland. Governance and proper planning of urban land development locations and procedures is a key measure to reduce future losses in crop production^{43, 56}. However, the governance qualities appear to be moderate to low due to the fragmented governance structures in many countries that will suffer from large crop production losses⁵⁷. The effective protection of croplands, therefore, calls for increasing efforts in educating policy makers and the development of more comprehensive governance regimes⁵⁸.”

References: Does this manuscript reference previous literature appropriately? If not, what references should be included or excluded?

There is a whole bunch of literature regarding modelling of SSP related scenarios that should be included.

Response:

Literature regarding modelling of five SSP scenarios has been added into reference:

“16. van Vuuren, D.P. et al. Energy, land-use and greenhouse gas emissions trajectories under a green growth paradigm. *Global Environmental Change* 42, 237-250 (2017).”

“17. Fricko, O. et al. The marker quantification of the Shared Socioeconomic Pathway 2: A middle-of-the-road scenario for the 21st century. *Global Environmental Change* 42, 251-267 (2017).”

“18. Fujimori, S. et al. SSP3: AIM implementation of Shared Socioeconomic Pathways. *Global Environmental Change* 42, 268-283 (2017).”

“19. Calvin, K. et al. The SSP4: A world of deepening inequality. *Global Environmental Change* 42, 284-296 (2017).”

“20. Kriegler, E. et al. Fossil-fueled development (SSP5): An energy and resource intensive scenario for the 21st century. *Global Environmental Change* 42, 297-315 (2017).”

Clarity and context:

I already pointed out that both the introduction and discussion need to be revised.

My overall recommendation is that the authors should substantially revise their manuscript before publication. Most importantly they have to add further simulation experiments to explore the model behaviour in more detail.

References:

Popp, A., Calvin, K., Fujimori, S., Havlik, P., Humpenöder, F., Stehfest, E., ... & Hasegawa, T. (2017). Land-use futures in the shared socio-economic pathways. *Global Environmental Change*, 42, 331-345.

Seto, K. C., Güneralp, B., & Hutyra, L. R. (2012). Global forecasts of urban expansion to 2030 and direct impacts on biodiversity and carbon pools. *Proceedings of the National Academy of Sciences*, 109(40), 16083-16088.

Response:

The suggested literatures have been added to the references:

“8. Seto, K.C., Guneralp, B. & Hutyra, L.R. Global forecasts of urban expansion to 2030 and direct impacts on biodiversity and carbon pools. *Proceedings of the National Academy of Sciences* 109, 16083-16088 (2012).”

“23. Popp, A. et al. Land-use futures in the shared socio-economic pathways. *Global Environmental Change* 42, 331-345 (2017).”

Reviewer #3 (Remarks to the Author):

This manuscript reports on an application of the Future Land-Use Simulation (FLUS) model at the global scale, using the shared socioeconomic pathways (SSPs) scenarios that are now adopted by the Coupled Model Intercomparison Project Phase 6 (CMIP6). Within the SSP scenario framework, the authors also report on related findings: urban land distribution and form, and environmental effects.

The novelty of this work lies in the global application of a land use change model at a 1km resolution that is consistent with the SSPs. This product will undoubtedly be of interest to the climate modeling community, but the authors miss a significant opportunity to learn and communicate about global urban land cover change trends and modeling challenges. As it currently stands, this manuscript reads as a technical exercise in modeling and GIS analysis, with little depth or critical interpretation. For these reasons, elaborated below, I recommend that this paper be rejected, but further work might justify a resubmission.

Central criticisms:

The authors present their modeling work largely as a technical exercise, without much critical discussion of how their modeling approach and assumptions will influence the results. Sohl et al. (2016), for example, demonstrate the low level of agreement among land use change models, even when they are informed by the same set of scenario assumptions. This indicates that the assumptions that are implicit with a particular modeling approach (or even within a particular modeling team) has a strong influence on the outcome. This is an issue that land change modelers need to consider and discuss when presenting their work. To point to a specific example in this manuscript, the authors state in the abstract that “a patch-based urban growth strategy is necessary...for the distribution of urban land-use to be consistent with Zipf’s law.” While the authors acknowledge that Zipf’s law is a primary guiding assumption, and they point to the relevant literature, they do not discuss the implications of this assumption – aside from the FoM comparisons presented in Table 1.

Response:

Urban systems feature the complexity of scaling, which can be described by using Zip’s law. Empirical evidences have demonstrated that cities in many countries follow Zip’s law (Jiang et al., 2015). Moreover, literature also reveals that the intra-urban scale distribution of urban land patches fits this law well (Fragkias et al., 2009).

“The distribution of population size for cities within a specific region has been found to follow the rank-size rule (Zipf’s law)¹⁵. The distribution of urban land cover patches in a specific region has also been found to follow the rank-size rule^{16,17}. Zipf’s law is the basic criterion we followed in the simulation by using a patch-based strategy, which is useful to yield better performances of urban simulation¹⁸⁻²⁰. The estimation of the rank-size distribution exponent (λ) is usually performed through a linear regression on log–log plots, as the slope of the regression line is the estimate of the exponent¹⁶. Comparing the logarithms of patch rank and patch size, researchers have found that the exponents remain stable over time^{16, 17}. This result indicates that larger patches tend to grow relatively faster than smaller patches; thus, the exponents can be

maintained in different periods. Therefore, we propose a patch-based urban growth strategy and apply it to the global urban simulation in this study, which can increase the potential for the development of large urban patches.”

“The implementation of a patch-based simulation strategy has successfully revealed the changes in urban systems in terms of the rank-size distributions of urban land patches (i.e., $\ln(\text{rank}_{\text{patch}}) = \lambda \ln(\text{size}_{\text{patch}})$) (Fig. 4).”:

Fig. 4 | The rank-size distributions of urban land patches in the representative regions in different periods. The LAM-L region consists of the low-income countries of Latin America, including Belize, Guatemala, Haiti, Honduras, and Nicaragua.

Similarly, there is no comprehensive discussion of error. Figure 1, for example, does not present errors bars around their estimates. Likewise, the use of FoM (Table 1) should be defined and interpreted more clearly. The authors state that FoM values “usually range from 10 percent to 30 percent” in other simulation studies, but readers should be given an indication of what the FoM would be for a “perfect fit.” This would allow readers to evaluate and contextualize the significance of overall model accuracy by region, and the significance of the contribution of the patch-based strategy.

Response:

We’ve added the uncertainty analysis for urban land demand estimation and spatial locations. Specifically, we forecast the urban land demand under the 95% confidence intervals for the panel

regression model. The results are shown in Fig. 1 which has been added in the revised manuscript.

Fig. 5 | Projections of urban land demand globally and in the representative macro regions for 2010–2100 under the SSP scenarios. The region LAM-L consists of the low-income countries of Latin America, including Belize, Guatemala, Haiti, Honduras, and Nicaragua. The shaded areas represent the 95% confidence intervals of the projected urban land demand.

For the spatial simulation, multiple experiments were conducted to identify the modeling uncertainty. We run our model for 100 times and overlap the results of these simulations. We assume that areas (in grids) with higher agreements of simulated growth among the 100 runs of simulations are more certain to be urbanized in the future. For instance, a grid that has been simulated as urban in 70 out of the 100 runs is assumed more likely to become urbanized than another grid that has been simulated as urban in, for example, 50 out of the 100 runs. These results are shown in Fig. 3 and have been added in the revised manuscript.

Fig. 3 | Uncertainty in the simulation results of three major metropolitan areas in 2100. This figure shows the likelihood of each grid to become urban, which is estimated by overlapping the results of 100 simulation runs. Grids with higher agreements of simulated growth among the 100 simulation runs are assumed more likely to become urbanized.

In the Supplementary Information we added a detailed introduction and definition about “Figure of Merit”:

“Conventional accuracy indicators, such as the overall accuracy and Kappa coefficient, are used to evaluate the simulation performances of both changed and persistently non-changed simulations. This may lead to the overestimation of accuracies when evaluating the simulation performance²¹. However, FoM avoids overestimation by only focusing on the parts of land where changes have taken place^{22, 23}. This index can be mathematically expressed as the ratio of the correct predicted change to the sum of the observed change and predicted change:

$$\text{FoM} = B / (A + B + C + D) \quad (6)$$

“where A represents the error area due to observed change predicted as persistence, B is the area of correctness due to observed change predicted as change, C denotes the area of error due to observed change predicted as change in the wrong category, and D denotes the area of error due to observed persistence predicted as change.”

In addition, in the Supplementary Information file, we further explain why the FoM values prove that our simulation accuracy is reliable:

“We use FoM, which is an indicator ranging from 0% to 100% that reflects the simulation accuracy by focusing only on the part of the land that has changed, to measure the performance of the simulation results for urban land cover change from 2000 to 2015^{22, 23}. Supplementary Table 1 shows the FoM values of the two methods for simulating global urban land cover change in 32 regions from 2000 to 2015. These methods, with and without the patch-based strategy, yield mean FoM values of 24.48% and 24.42% for the 32 regions, respectively. Our results are similar to those of other case studies on land cover change modelling. Previous comparative analysis has demonstrated that the common range of FoM values is between 10% and 30% for existing land cover change models^{12, 19}. A recent global land cover simulation conducted by Li et

al.¹² also reported similar results of FoM values, which ranged from 10% to 29%, with a mean value of 19%. Additionally, empirical studies have found that the value of FoM is strongly affected by the relative amount of observed net change²². Small FoM values were usually found in cases with a small amount of observed net change. For instance, Pontius et al.²² revealed that the FoM value may decrease to less than 8% if the observed net change accounts for less than 2% of the total land area. In our case, however, the observed urban land growth only accounts for 0.06% of the global terrestrial area, while the FoM values of our model exceed 24%. In this sense, the performance of our model is reliable, as indicated by the relatively higher FoM values, given a small amount of observed net change.”

The above comment also applies to what seems to be a key input into the model, the probability of occurrence surfaces. The input variables (listed in Supplementary Table 2) will have their own error sources – temporally, for example, they range from 2000 to 2015, so forecasts out to 2100 will compound errors related to temporal mismatch. As an aside, I do not think the authors provide enough background to reproduce this layer, noting in the methods section that the probability of occurrence surfaces are estimated “with the help of our own software.”

Response:

In the discussion section we discussed the error sources in our model, including errors of the input variables.

“There are several limitations in our projections. First, we do not address the potential errors in future urban land expansion that are caused by the misclassifications in historical urban land maps, which were acquired from the GHSL dataset. Second, we hold the spatial driving factors of urban land expansion constant in the scenario projections, which is mainly because of the difficulty in the reliable prediction of, for example, changes in urban infrastructure. Third, we do not consider the impacts of future climate change on land-use change. This limitation can be alleviated by applying the projections under the integrated scenarios of RCP-SSP, which take into account the feedbacks of global climate changes.”

In the Supplementary Information file, we have improved the description of how to estimate the probability-of-occurrence surfaces:

“The estimation of the ‘probability-of-occurrence’ is based on the ANN method with a set of spatial variables. These spatial variables are selected mainly according to the findings of previous studies on urban land expansion simulation¹¹⁻¹³. They include population, GDP, distance to the city centre, distance to the road network, distance to the airport, elevation, slope and the water resource condition, which collectively cover the major socioeconomic and natural factors affecting urban land expansion¹¹⁻¹³. To take into account the ecological characteristics, we also acquire the ecoregion types from a world ecoregions map (Supplementary Table 2) and use them as nominal features in the ANN training. The corresponding data sources of all these spatial variables are provided in Supplementary Table 2. All these variables are processed to have a spatial resolution of 1 km with a spatial reference of *WGS_1984_Cylindrical_Equal_Area*.

The estimation of the ‘probability-of-occurrence’ is carried out separately for each of the 32 macro regions. Specifically, for each macro region, a sample is collected using a stratified random

sampling approach with an equal proportion of 30% for urban and non-urban grids, respectively. This sample is used as a training dataset to train an ANN that yields the corresponding classification probabilities for 'urban' or 'non-urban'. The training procedure is based on the traditional back-propagation (BP) approach¹⁴. After the training procedure, for a non-urban grid i of land cover type k , its value of 'probability-of-occurrence' $P_{i,k}$ is represented using the 'urban' classification probability obtained from a trained ANN."

Another methodological choice that was made with little discussion is the choice of 32 modeling regions. While the authors may be constrained to this choice by the SSPs, it is worth discussing whether or not these regions make sense from an urban modeling perspective.

Response:

In the revised paper, we have explained the basis for the division of the 32 regions:

"We follow the region definitions in the SSP database (<https://tntcat.iiasa.ac.at/SspDb>), which aggregate the world's countries into 32 macro regions according to the conditions of geographic location and income level (i.e., high-income, mid-income and low-income levels)."

When presenting results of the impacts of urban expansion on the environment, the authors make some very broad statements that may not be justified based on a simple analysis of land use change trajectories under the SSP scenarios. Statements such as "it is also apparent that excessive urban growth may lead to serious issues such as excessive energy consumption," "slow urban growth reflects the stagnation of the development of socio-economic factors...a particularly detrimental impact on developing countries," "taking the green road (SSP1) can maintain a city's development with enough impetus while avoiding chaos" are unfounded given the data the authors are referencing. A more nuanced discussion here is needed.

Response:

We have removed such statements to avoid confusion. In the revised manuscript, we discuss the implications that strictly come from our findings. For example, we have enhanced the scenario analysis of future urban growth and its impacts:

"Based on the urban land projections, we have identified the pressure caused by future urban shrinkage for different regions across the world. Urban shrinkage occurrences largely correspond to low birth rates and population ageing. Typical regions that will suffer from substantial pressure from urban shrinkage mainly include China and many other Asian countries. North America, South America and western Europe experience great pressure from urban shrinkage mainly in scenarios SSP3 and SSP4. These results have important implications for urban planning and management. In rapidly urbanizing countries, such as China, urban planning is usually prepared with the expectation of growth, with little or no awareness of the ongoing/upcoming challenges caused by urban shrinkage⁴¹. A direct consequence of urban shrinkage is the difficulty of sustaining the maintenance of infrastructure, which was originally built to serve a larger population, leading to declines in livelihoods and quality of life in cities. Abandonments and social disorder are also potential outcomes raised by urban shrinkage that threaten public health and safety in cities^{46, 47}. Moreover, from an economic point of view, urban shrinkage may cause economic recession because of the negative impacts on employment, industrial production and

property value⁴⁸⁻⁵⁰.”

“Despite the potential urban shrinkage in some regions of the world, global urban land is expected to experience rapid growth and modify the global landscape before the 2040s. Our projections demonstrate that most future urban land expansion occurs over croplands (50-63%) and forests (30-44%). The losses in croplands and forests will have profound impacts on ecosystem services, such as carbon sequestration, habitat provision and food supply to human societies⁵¹⁻⁵⁴. Our results show that the crop production losses induced by urban expansion will affect the annual food needs of approximately 122-1389 million people. More importantly, the crop production losses (approximately 1-4%) are disproportional to the cropland land losses (approximately 1%). Previous studies have revealed that a 2 °C global temperature increase will cause 8-14% of global maize production losses⁵⁵. In comparison, our projections suggest that urban land expansion alone will lead to 1-3% of global maize production loss, indicating the profound impacts of urban expansion on global crop production. These results imply the critical importance of conserving fertile and productive cropland. Governance and proper planning of urban land development locations and procedures is a key measure to reduce future losses in crop production^{43,56}. However, the governance qualities appear to be moderate to low due to the fragmented governance structures in many countries that will suffer from large crop production losses⁵⁷. The effective protection of croplands, therefore, calls for increasing efforts in educating policy makers and the development of more comprehensive governance regimes⁵⁸.”

The authors frequently use imprecise or unclear language. For example, the use of the term “urban land use” is problematic, since there is no differentiation of “uses.” “Urban land cover” may be a more appropriate phrase, although ultimately the authors should define what “urban” encompasses. “Urbanization” is also used in the methods section, and I infer this to mean the proportion of the population living in an urban area, but it is not defined. Other confusing or poorly defined terms and concepts include: polarized urban form, dispersive patch size distribution, dynamical urban land-use change (vs. dynamic, which is also used in similar contexts), panel data regression method.

Response:

The language of the manuscript has been revised and improved. We also used a professional language editing service to further refine the writing of this manuscript. In the revised manuscript, we have replaced the term “urban land use” as “urban land cover”:

“Here ‘urban land’ refers to artificial cover and paved surfaces.”

The method section has been rewritten and the term “urbanization” has been removed. We define the term “urbanization rate” when it first appears:

“% of urban population to total population”.

The expressions “polarized urban form” and “dispersive patch size distribution” have been removed. We have replaced the statement “dynamical urban land-use change” as “urban land change”. We have defined “panel data regression” when it first appears:

“Here panel data regression refers to a regression method that simultaneously uses

cross-sectional and longitudinal data.”

Finally, the authors should explicitly state whether they are able to incorporate climate change-land use/cover change feedbacks. I’m guessing this is not the case, but it is worth stating and noting it as a potential limitation – and an avenue for future research.

Response:

In the revised paper, we have discussed the limitations and potential of this study in relation to climate change:

“There are several limitations in our projections....Third, we do not consider the impacts of future climate change on land-use change. This limitation can be alleviated by applying the projections under the integrated scenarios of RCP-SSP, which take into account the feedbacks of global climate changes.”

I realize that many of the concerns I expressed regarding methods are answered in some of the background publications the authors reference. They are clearly building on years of research and development around the FLUS model. However, this manuscript currently does not stand alone, and in my opinion the authors should improve the communication of their approach and results such that an informed reader can reasonably follow along without delving into the more detailed treatments.

Response:

We have revised the description of our method. In particular, “Estimation of urban land demand” in the Method section, and “Future Land-Use Simulation (FLUS) model” in the Supplementary Information have been explained in more details.

Reference: Sohl, TL, MC Wimberly, VC Radeloff, DM Theobald, and BM Sleeter. Divergent projections of future land use in the United States arising from different models and scenarios. *Ecological Modelling* 337(2016) 281-297. <http://dx.doi.org/10.1016/j.ecolmodel.2016.07.016>

Response: The suggested literatures have been added to the references:

“11. Sohl, T.L., Wimberly, M.C., Radeloff, V.C., Theobald, D.M. & Sleeter, B.M. Divergent projections of future land use in the United States arising from different models and scenarios. *ECOL MODEL* 337, 281-297 (2016).”

** See Nature Research’s author and referees’ website at www.nature.com/authors for information about policies, services and author benefits

This email has been sent through the Springer Nature Tracking System NY-610A-NPG&MTS

Confidentiality Statement:

This e-mail is confidential and subject to copyright. Any unauthorised use or disclosure of its contents is prohibited. If you have received this email in error please notify our Manuscript Tracking System Helpdesk team at <http://platformsupport.nature.com> .

Details of the confidentiality and pre-publicity policy may be found here <http://www.nature.com/authors/policies/confidentiality.html>

Privacy Policy | Update Profile

Reviewers' comments:

Reviewer #1 (Remarks to the Author):

Rereview of NCOMMS-18-19429A

The authors have made a good faith effort to respond to my previous comments. I think the new title, as well as the focus on the results of their urban growth model, rather than analyses of soil carbon impacts, has strengthened the paper. I particularly like the inclusion of uncertainty analyses.

Also I approve of the changes in the methodology around shrinkage (i.e., not assuming a reduction in urban area would accompany a reduction in urban population).

They were also very thorough and thoughtful in responding to my suggestions on text edits, for which I am appreciative.

Bigger criticisms:

1. I would insist that all urban growth scenario projections and data be put into a permanent online data repository, that will generate a DOI for them. While it is great the authors have set up a website, my experience has been that it is hard for researchers to maintain websites beyond 3-5 years, and for a journal publication ideally the data should be put in a permanent repository (there are lots to choose from).

2. This article should cite Zhou et al.'s recent paper in NATURE SCIENTIFIC DATA: Zhou, Y., Varquez, A. C. G. & Kanda, M. High-resolution global urban growth projection based on multiple applications of the SLEUTH urban growth model. Scientific Data 6, 34, doi:10.1038/s41597-019-0048-z (2019).

Moreover, since this article is aiming to be in NATURE COMMUNICATIONS (a high tier journal), it needs to make the case for why its urban growth scenarios are better/novel compared with the Zhou scenarios. They will need to convince readers and the editor that their method is better than Zhou's, or that there is at least scientific merit to having both sets of urbanization scenarios published.

3. I think that the projected urban growth under the SSPs should be contrasted with other land-use changes under the SSPs (drawing from other published papers forecasting land-use change under the SSPs). Are the SSPs with lots of agricultural expansion, for instance, the same ones with lots of urban expansion? Or are their contrasting patterns?

4. Rather than talking about "urban shrinkage", I would suggest you just talk about "urban population decline". This makes clear that you are talking about a demographic trend, with unclear effects on urban area.

Text suggestions:

Abstract- Suggest rather than "urban shrinkage", that you say "urban population decline". This avoids making any assumptions in your language that urban area will decline (which I think history suggests is unlikely).

Line 13-14: "... is expected to occur on current cropland" would be perhaps more clear. "Global crop production will decline by approximately 1-4%, corresponding to the annual food needs of 122-1389 million people" = This sentence is surprising to readers at this point, since it is unclear how a decline of 4% in food production could somehow equal the food needs of close to 20% of the global population! I will check the methods section to see how this is calculated, but just wanted to say that for readers first encountering this finding in the abstract, this finding seems counterintuitive.

Body text:

Line 91- Here, it may be helpful to contrast the FLUS model with the SLEUTH model than Zhou et al. used. Why is FLUS better/more accurate?

Figures:

Figure 1: I find it interesting that the uncertainty is so much less for US predictions than for China or LAM. Might be worth discussing this in the text (perhaps this is in there and I missed it, my apologies if that is the case).

Reviewer #2 (Remarks to the Author):

The authors have significantly improved their paper. Nevertheless there are several issues that need to be clarified before publication.

My major concern is related to the comment in my first review where I asked the authors for further experiments such as sensitivity and uncertainty analyses to test their model. Such an analysis has been done only for input data regarding the area demand per person so far which in my opinion is not sufficient when a new methodology is presented. The uncertainties shown in the spatial distribution are solely due to the stochastic character of allocation model and do not consider parameter, data or structural uncertainties in a strict sense. This shortcoming should at least be addressed in the discussion.

Moreover a critical reflection of the modelling approach is still missing. The authors only compare their land-cover product (maps) with other products. A discussion of pros and cons of the model itself and how it differs from other urbanization models has not been done properly. In this context also questions of cross-scale dependencies should be addressed (at least) in the discussion. For example drivers such as demand for urban area are provided only for different world regions while policies for urban development are operating on country-level.

The idea to investigate cropland losses from urbanization is good, but to draw conclusions on crop production losses and affected people is quite misleading. First, there is no interaction between urbanization and cropland development modelled, e.g. cropland conversion to urban area might trigger expansion of cropland elsewhere. Second, the authors did not consider yield changes due to changes in agricultural management practices which might compensate the projected losses. Third, the number of affected people will strongly vary between world regions due to access to food and different diets.

The discussion is still imbalanced and should address the modelled urbanization trends and patterns in greater regional detail. Again, I would expect a more critical reflection of the consequences of these land-use changes not only on cropland losses but also on societies.

Reviewers' comments:

Reviewer #1 (Remarks to the Author):

Rereview of NCOMMS-18-19429A

The authors have made a good faith effort to respond to my previous comments. I think the new title, as well as the focus on the results of their urban growth model, rather than analyses of soil carbon impacts, has strengthened the paper. I particularly like the inclusion of uncertainty analyses.

Also I approve of the changes in the methodology around shrinkage (i.e., not assuming a reduction in urban area would accompany a reduction in urban population).

They were also very thorough and thoughtful in responding to my suggestions on text edits, for which I am appreciative.

Bigger criticisms:

1. I would insist that all urban growth scenario projections and data be put into a permanent online data repository, that will generate a DOI for them. While it is great the authors have set up a website, my experience has been that it is hard for researchers to maintain websites beyond 3-5 years, and for a journal publication ideally the data should be put in a permanent repository (there are lots to choose from).

Response:

We have put our urban growth scenario projections and data into a permanent online data repository, PANGAEA, with the DOI: <https://doi.pangaea.de/10.1594/PANGAEA.905890>, as well as our website. Now, this dataset in PANGAEA has not yet been released until the paper is published. The reviewers can be temporarily accessed through this link: <https://www.pangaea.de/tok/8b45b7cf6c31696dcde8f6051657531f57b4203f>.

“Here, we present spatially explicit projections of global urban land expansion from 2015 to 2100 and discuss its impacts. The open-access dataset of these projections is available via <https://doi.pangaea.de/10.1594/PANGAEA.905890> OR <http://www.geosimulation.cn/GlobalSSPsUrbanProduct.html>.”

2. This article should cite Zhou et al.'s recent paper in NATURE SCIENTIFIC DATA:

Zhou, Y., Varquez, A. C. G. & Kanda, M. High-resolution global urban growth projection based on multiple applications of the SLEUTH urban growth model. Scientific Data 6, 34, doi:10.1038/s41597-019-0048-z (2019).

Moreover, since this article is aiming to be in NATURE COMMUNICATIONS (a high tier journal), it needs to make the case for why its urban growth scenarios are better/novel compared with the Zhou scenarios. They will need to convince readers and the editor that their method is better than Zhou's, or that there is at least scientific merit to having both sets of urbanization scenarios

published.

Response:

We have included Zhou's research in the revised manuscript. The major difference is that Zhou's research only provides a single scenario based on the historical trajectory. They estimate the future demand without considering the potential pathways and uncertainties of future socio-economic factors. The use of a single scenario based on the historical trajectory may not be consistent with the recent IPCC framework, hindering the further application of urban growth projections in climate and environmental change studies. Our simulations are based on multiple scenarios of SSPs, which "describe how global society, demographics, and economics will change in the coming century concerning policy assumptions and the socio-economic narrative (e.g., energy demand and supply and technological change)^{13, 14.}" We've included the following information in the revised Introduction:

"However, some models only provide a single scenario based on the historical trajectory in the simulation of future global urban growth.¹² The simulation based on this single scenario could hinder the applications in climate and environmental change studies. In our study, we carry out scenario simulations based on the shared socioeconomic pathways (SSPs), mainly because SSPs provide a more comprehensive framework by considering the potential pathways and uncertainties of future socio-economic factors."

3. I think that the projected urban growth under the SSPs should be contrasted with other land-use changes under the SSPs (drawing from other published papers forecasting land-use change under the SSPs). Are the SSPs with lots of agricultural expansion, for instance, the same ones with lots of urban expansion? Or are their contrasting patterns?

Response:

In the revised paper, we supplement the comparisons with other products:

"Our projections are comparable to existing global urban land projections, except that our results have a much higher spatial resolution. We choose three representative products for the comparison. The first one is the 2030 global urban expansion product based on a single UN scenario at a 5-km resolution, which is created by Seto et al.⁸ The second one is the 2050 global urban growth projection based on a historical trajectory with a 1-km resolution, which is created by Zhou et al.¹² The third one is the 0.25-degree (about 27-km on the equator) LUH2 dataset that follows the assumptions of SSPs²⁷. Since there is no historical trajectory in the SSPs, we select the results of the middle pathway (SSP2) to compare with the single scenario projections mentioned above. The comparison is implemented in different regions, as shown in Fig. 5. The distribution of urban land areas is similar among the selected projections. Our results are most consistent with the urban land areas in LUH2, as indicated by a significant Pearson correlation coefficient of 0.93. The consistency is also high between our projections and Seto's results (Pearson correlation coefficient = 0.82). Our results show partial agreements with Zhou's results, as indicated by a relatively lower but significant Pearson correlation coefficient of 0.56. The spatial agreements between our projections and the three selected products are shown in Supplementary Figure 19. Some evident spatial differences between our results and Seto's/Zhou's results can be found in

regions such as the North China Plain. It is mainly because their projections do not consider the population decline trend in these regions, which can cause urban land growth to become stagnant⁴⁰.”

Fig. 5 | Comparison of urban land area of 2030 in different regions by Seto's, Zhou's, LUH2 and our model.

Supplementary Figure 19 | **The difference between our product and other global urban land products in 2030. (a) Ours in SSP2 and Seto's; (b) Ours in SSP2 and Zhou's; (c) Ours and LUH2's in SSP2.** The comparisons are based on the urban land proportions in a 0.25 degree grid. The levels are divided by 10% of the difference in the urban land

proportions. That is, "Level 0" means the difference in the urban land proportion between two products is less than 10%, "Level 1" means less than 20%, and so on.

Meanwhile, we supplement the analysis of the interaction between urban land and cropland, considering the land productivities in different scenarios:

"Despite the potential decline of urban population in some regions of the world, global urban land is expected to experience rapid growth before the 2040s. Our projections have demonstrated that most future urban land expansion occurs over croplands (50-63%) and forests (30-44%), causing the losses of food production and supply. To offset these losses, however, reclaiming cropland remains as a most immediate way, because land productivity may be difficult to improve substantially in the foreseeing future. The losses in croplands and forests will also have profound impacts on ecosystem services, such as carbon sequestration, habitat provision and food supply to human societies⁵⁵⁻⁵⁸."

"Note that in this study we only account the direct impacts of urban expansion on other land types. To gain a clearer understanding, we complement our results with those obtained from Popp et al.²⁴, which provide scenario projections of cropland expansion on other land use types. Their results suggest that the SSP3 scenario features the largest amount of cropland expansion (approximately 700 million ha from 2005 to 2100), while the SSP1 scenario has the smallest increase of cropland area (approximately 20 million ha from 2005 to 2100). For SSP2, SSP4 and SSP5, the cropland areas are expected to increase between 250 million ha and 400 million ha from 2005 to 2100. Our scenario projections, which mainly focus on the direct impacts of urban expansion on cropland, reveal that SSP5 has the largest cropland loss due to urban growth (37.6 million ha), while SSP3 has the smallest loss (22.0 million ha). The estimated cropland losses in other SSP scenarios vary slightly from 22.4 million ha to 24.9 million ha. Therefore, the combination of our results and Popp's reveals the contrasting patterns of urban land and cropland expansion. In the scenarios with more cropland losses caused by urban expansion, there are relatively small increases of cropland, and vice versa."

4. Rather than talking about "urban shrinkage", I would suggest you just talk about "urban population decline". This makes clear that you are talking about a demographic trend, with unclear effects on urban area.

Response:

In the revised paper, we have replaced "urban shrinkage" with "urban population decline" according to the referee's advice.

"Noticeable declines in urban land demand are observed in our scenario projections, mainly owing to the decrease in population. A typical example is China (Fig. 1b), in which urban land demand sharply decreases in all scenarios after the 2040s/2050s due to the declining population (Fig. 6). Urban population decline does not necessarily lead to a massive land conversion from urban to non-urban areas, although some individual cases of green recovery from urban land can

be found in contemporary Germany at the block scale⁴¹. Rather, one possible consequence of urban population decline is the abandonments of the already built-up areas. This has been occurring in some countries around world^{42,43}, even in the rapid urbanizing country of China⁴⁴.”

“However, to address the decline in urban land demand in the spatial simulations, we assume that the land conversion of non-urban land to urban land is irreversible and that the massive conversion of urban land to non-urban land is not allowed. For a certain region, if its estimated urban land demand is smaller than its total area of the already built-up, then no changes are simulated for this region, and the spatial extent of existing urban land also remains unchanged. Subsequently, for regions experiencing a decline in urban land demand, we assess their pressure on urban population decline, which is defined as the percentage of the area of “urban land surplus” (unnecessary urban land compared to urban population) over the area of existing urban land, ranging from 0 to 100%. The results (Fig. 7) suggest that China will no doubt face the most severe pressure of urban population decline compared to other countries around the world. South America and almost all Asian areas will encounter substantial pressure of urban population decline in the SSP1 and SSP4 scenarios, while European countries and North America will face increased pressure of urban population decline in the SSP3 scenario.”

Text suggestions:

Abstract- Suggest rather than "urban shrinkage", that you say "urban population decline". This avoids making any assumptions in your language that urban area will decline (which I think history suggests is unlikely).

Response:

We have replaced "urban shrinkage" with "urban population decline" in the abstract:

“Despite its small land coverage, urban land and its expansion have exhibited profound impacts on global environments. Here, we present the scenario projections of global urban land expansion under the framework of the shared socioeconomic pathways (SSPs). Our projections feature a fine spatial resolution of 1 km to preserve spatial details. The projections reveal that although global urban land continues to expand rapidly before the 2040s, China and many other Asian countries are expected to encounter substantial pressure from urban population decline after the 2050s. Approximately 50-63% of the newly expanded urban land is expected to occur on current croplands. Global crop production will decline by approximately 1-4%, corresponding to the annual food needs (for a certain crop) of 122-1389 million people. These findings stress the importance of governing urban land development as a key measure to mitigate its negative impacts on food production.”

Line 13-14: "... is expected to occur on current cropland" would be perhaps more clear.

Response:

We have modified the corresponding sentence: “Approximately 50-63% of the newly expanded urban land is expected to occur on current croplands.”

"Global crop production will decline by approximately 1-4%, corresponding to the annual food needs of 122-1389 million people" = This sentence is surprising to readers at this point, since it is

unclear how a decline of 4% in food production could somehow equal the food needs of close to 20% of the global population! I will check the methods section to see how this is calculated, but just wanted to say that for readers first encountering this finding in the abstract, this finding seems counterintuitive.

Response:

We have described the method for estimating the affected population, and explained the estimated results with more details. In addition, the main reason for the large population affected by the loss in maize production is that the maize used for food supply (the global annual per capita food supply of maize) is much less than that used for feeding:

“The diet structures of people vary largely from one place to another. To estimate the number of people affected by the global crop production losses, we use the global average per capita food supply of rice, wheat, maize, potatoes and vegetables as the basis for the estimation. These statistics can represent the overall situation of food supply worldwide and hence are adequate to apply in our analysis (<http://www.fao.org/faostat/en/>).”

“The estimated number of people affected by the losses of crop production vary widely from 122 million to 1389 million, depending on which food consumption statistics are used (Table 2). Note that these results are not the numbers of people suffering from starving, but the maximum numbers of people affected by the production losses of a single food crop type, if these losses cannot be compensated with productions of other food crops. In addition, we did not consider yield changes due to changes in agricultural management practices which might compensate the projected losses.”

Body text:

Line 91- Here, it may be helpful to contrast the FLUS model with the SLEUTH model than Zhou et al. used. Why is FLUS better/more accurate?

Response:

The SLEUTH model can only input five spatial driving layers (slope, excluded, urban, transportation, hillshade) for the urban land simulation, which corresponds to the five coefficients that control the behaviour of the system (Clarke 1998; Zhou et al. 2019). This mechanism constrains the model to consider more other spatial driving factors. However, the SSPs describe how global society, demographics, and economics will change in the coming century concerning policy assumptions and the socio-economic narratives (e.g., energy demand and supply and technological change). Compared to the SLEUTH model, the SSPs-FLUS model can conveniently consider the socio-economic and physical spatial driving factors, and facilitate the exploration of future land use change scenarios.

Moreover, Zhou's research only provides a single scenario based on the historical trajectory. They derive the rate of urban expansion from the past period, and use it to estimate the future demand without considering the potential pathways and uncertainties of future socio-economic factors. The use of a single scenario based on the historical trajectory may not be consistent with

the recent IPCC framework, hindering the further application of urban growth projections in climate and environmental change studies. Our simulations are based on multiple scenarios of SSPs, which can interface with the IPCC framework.

We presented the difference between our model and the SLEUTH model that Zhou et al. used in section “Comparison with other urbanization models” in the supplementary information:

“In Zhou et al.’s research²⁵, they used SLEUTH model to simulate future land changes according to the historical trends captured from a series of historical land maps. However, their simulation only used a single scenario based on the historical trajectory. This simulation had not considered the potential pathways and uncertainties of future socio-economic factors and thus was incompatible with the recent IPCC framework. Moreover, the SLEUTH model can only input five spatial driving layers (slope, excluded, urban, transportation, hillshade) for the urban land simulation, which corresponds to the five coefficients that control the behaviour of the system^{25, 27}. These coefficients provide convenience for interpreting the mechanism of the model, but limit the consideration of more factors in the simulation process.”

“Our projections are generated using FLUS model coupled with the latest SSPs, which describe potential pathways in the coming century concerning policy assumptions and the socio-economic narrative, being consistent with the recent IPCC framework. These projections thus provide better potential in supporting the research in related disciplines, such as ecological protection, water security, urban climate and global climate change.”

Figures:

Figure 1: I find it interesting that the uncertainty is so much less for US predictions than for China or LAM. Might be worth discussing this in the text (perhaps this is in there and I missed it, my apologies if that is the case).

Response:

In the revised paper, we supplemented the estimated coefficients and their uncertainties of using the panel data regression:

“Compared to the other two regions, the USA has the least uncertainty because it achieves a better fit in the panel data regression.”

“The estimated coefficients have passed a significance level of 0.01. The values of β_0 in each region reflect the heterogeneity between different regions⁴⁹ (detailed results of the estimated coefficients are shown in Supplementary Table 2).”

In the panel data regression, all the regions share the values of coefficients β_1 and β_2 , but each region has a different value of β_0 (Intercept). Here the β_0 (Intercept) reflects the heterogeneity of different regions. The resulting β_0 for USA is relatively large, but the resulting standard deviation is small relative to the β_0 , and hence makes the uncertainty of the urban land demand small for the USA.

Supplementary Table 1 | The results of the panel data regression results for per capita urban

land demand (unit: km² / million persons)

Coefficients	Estimate	Std. Error	t value	Pr(> t)
β_1 (GDPC) / (\$/person)	0.0006	0.00	2.96	<0.005
β_2 (PU) / (%)	0.9154	0.24	3.84	<0.001
β_0 (Intercept)				
ANUZ	182.80	20.99	8.71	<0.001
BRA	32.90	12.16	-12.33	<0.001
CAN	101.23	11.69	-6.98	<0.001
CAS	54.70	15.46	-8.28	<0.001
CHN	40.60	16.39	-8.68	<0.001
EEU	28.20	14.92	-10.36	<0.001
EEU-FSU	48.70	13.44	-9.98	<0.001
EFTA	-21.70	12.59	-16.24	<0.001
EU12-H	39.10	12.75	-11.27	<0.001
EU12-M	14.50	13.95	-12.06	<0.001
EU15	38.20	11.85	-12.20	<0.001
IDN	96.84	15.99	-5.38	<0.001
IND	28.20	17.54	-8.82	<0.001
JPN	84.64	11.61	-8.46	<0.001
KOR	18.80	12.08	-13.58	<0.001

LAM-L	14.70	14.73	-11.41	<0.001
LAM-M	19.40	12.28	-13.31	<0.001
MEA-H	54.90	11.71	-10.92	<0.001
MEA-M	19.20	13.48	-12.14	<0.001
MEX	35.80	12.23	-12.02	<0.001
NAF	55.80	14.25	-8.91	<0.001
OAS-CPA	64.50	18.26	-6.48	<0.001
OAS-L	33.20	17.36	-8.62	<0.001
OAS-M	45.10	15.71	-8.76	<0.001
PAK	28.50	16.79	-9.19	<0.001
RUS	41.90	13.05	-10.80	<0.001
SAF	103.21	13.49	-5.90	<0.001
SSA-L	28.30	17.45	-8.86	<0.001
SSA-M	47.40	15.50	-8.74	<0.001
TUR	15.00	13.06	-12.85	<0.001
USA	136.71	11.89	-3.88	<0.001

Reviewer #2 (Remarks to the Author):

The authors have significantly improved their paper. Nevertheless there are several issues that need to be clarified before publication.

My major concern is related to the comment in my first review where I asked the authors for further experiments such as sensitivity and uncertainty analyses to test their model. Such an analysis has been done only for input data regarding the area demand per person so far which in my opinion is not sufficient when a new methodology is presented. The uncertainties shown in the spatial distribution are solely due to the stochastic character of allocation model and do not consider parameter, data or structural uncertainties in a strict sense. This shortcoming should at

least be addressed in the discussion.

Response:

In the revised paper, we have supplemented the discussion of uncertainties of our dataset in the discussion section:

“The uncertainty of modelling can be divided into stochastic uncertainty, parameter uncertainty, heterogeneity and structural uncertainty.⁴⁹ We quantified the parameter uncertainties in the projections of urban land demand (Supplementary Table 2). In order to deal with the heterogeneity, we assigned different intercepts to different regions in the panel data regression (Supplementary Table 2), implementing spatial simulations in different regions separately. We then performed 100 spatial simulations to understand the stochastic uncertainty of the model. To address the structural uncertainty⁴⁹, we compared our products with three other products using different models (Fig. 5 and Supplementary Figure 19).”

“Our projections are comparable to existing global urban land projections, except that our results have a much higher spatial resolution. We choose three representative products for the comparison. The first one is the 2030 global urban expansion product based on a single UN scenario at a 5-km resolution, which is created by Seto et al.⁸ The second one is the 2050 global urban growth projection based on a historical trajectory with a 1-km resolution, which is created by Zhou et al.¹² The third one is the 0.25-degree (about 27-km on the equator) LUH2 dataset that follows the assumptions of SSPs²⁷. Since there is no historical trajectory in the SSPs, we select the results of the middle pathway (SSP2) to compare with the single scenario projections mentioned above. The comparison is implemented in different regions, as shown in Fig. 5. The distribution of urban land areas is similar among the selected projections. Our results are most consistent with the urban land areas in LUH2, as indicated by a significant Pearson correlation coefficient of 0.93. The consistency is also high between our projections and Seto's results (Pearson correlation coefficient = 0.82). Our results show partial agreements with Zhou's results, as indicated by a relatively lower but significant Pearson correlation coefficient of 0.56. The spatial agreements between our projections and the three selected products are shown in Supplementary Figure 19. Some evident spatial differences between our results and Seto's/Zhou's results can be found in regions such as the North China Plain. It is mainly because their projections do not consider the population decline trend in these regions, which can cause urban land growth to become stagnant⁴⁰.”

Supplementary Table 2 | The results of the panel data regression results for per capita urban

land demand (unit: km² / million persons)

Coefficients	Estimate	Std. Error	t value	Pr(> t)
β_1 (GDPC) /	0.0006	0.00	2.96	<0.005

(\$/person)				
β_2 (PU) / (%)	0.9154	0.24	3.84	<0.001
β_0 (Intercept)				
ANUZ	182.80	20.99	8.71	<0.001
BRA	32.90	12.16	-12.33	<0.001
CAN	101.23	11.69	-6.98	<0.001
CAS	54.70	15.46	-8.28	<0.001
CHN	40.60	16.39	-8.68	<0.001
EEU	28.20	14.92	-10.36	<0.001
EEU-FSU	48.70	13.44	-9.98	<0.001
EFTA	-21.70	12.59	-16.24	<0.001
EU12-H	39.10	12.75	-11.27	<0.001
EU12-M	14.50	13.95	-12.06	<0.001
EU15	38.20	11.85	-12.20	<0.001
IDN	96.84	15.99	-5.38	<0.001
IND	28.20	17.54	-8.82	<0.001
JPN	84.64	11.61	-8.46	<0.001
KOR	18.80	12.08	-13.58	<0.001
LAM-L	14.70	14.73	-11.41	<0.001
LAM-M	19.40	12.28	-13.31	<0.001
MEA-H	54.90	11.71	-10.92	<0.001
MEA-M	19.20	13.48	-12.14	<0.001

MEX	35.80	12.23	-12.02	<0.001
NAF	55.80	14.25	-8.91	<0.001
OAS-CPA	64.50	18.26	-6.48	<0.001
OAS-L	33.20	17.36	-8.62	<0.001
OAS-M	45.10	15.71	-8.76	<0.001
PAK	28.50	16.79	-9.19	<0.001
RUS	41.90	13.05	-10.80	<0.001
SAF	103.21	13.49	-5.90	<0.001
SSA-L	28.30	17.45	-8.86	<0.001
SSA-M	47.40	15.50	-8.74	<0.001
TUR	15.00	13.06	-12.85	<0.001
USA	136.71	11.89	-3.88	<0.001

Fig. 5 | Comparison of urban land area of 2030 in different regions by Seto's, Zhou's, LUH2

and our model.

Supplementary Figure 19 | **The difference between our product and other global urban land products in 2030. (a) Ours in SSP2 and Seto's; (b) Ours in SSP2 and Zhou's; (c) Ours and LUH2's in SSP2.** The comparisons are based on the urban land proportions in a 0.25 degree grid. The level are divided by 10% of the difference in the urban land proportions. That is, "Level 0"

means the difference in the urban land proportion between two products is less than 10%, “Level 1” means less than 20%, and so on.

Moreover a critical reflection of the modelling approach is still missing. The authors only compare their land-cover product (maps) with other products. A discussion of pros and cons of the model itself and how it differs from other urbanization models has not been done properly. In this context also questions of cross-scale dependencies should be addressed (at least) in the discussion. For example drivers such as demand for urban area are provided only for different world regions while policies for urban development are operating on country-level.

Response:

In section “Comparison with other urbanization models” in the supplementary information, we have added a comparison of the models which are used to create the urban land products mentioned in our paper:

“Existing global future urban land products are mainly based on the scenarios including single forecasting scenario^{14, 25} and universal climate scenarios²⁶. Models used for these projections include SLEUTH²⁵, URBANMOD (a modified version of the GEOMOD model)¹⁴ and GGLM²⁶. All these models except SLEUTH project future land changes based on a single set of initial land map.”

“In Zhou et al.'s research²⁵, they used SLEUTH model to simulate future land changes according to the historical trends captured from a series of historical land maps. However, their simulation only used a single scenario based on the historical trajectory. This simulation had not considered the potential pathways and uncertainties of future socio-economic factors and thus was incompatible with the recent IPCC framework. Moreover, the SLEUTH model can only input five spatial driving layers (slope, excluded, urban, transportation, hillshade) for the urban land simulation, which corresponds to the five coefficients that control the behaviour of the system^{25, 27}. These coefficients provide convenience for interpreting the mechanism of the model, but limit the consideration of more factors in the simulation process.”

“The URBANMOD model, which is a modified version of the GEOMOD model, was used by Seto et al.¹⁴ to predict global urban land change by 2030. The single scenario formulated in their research is according to the projections of future socio-economic development made by the United Nations. In the original GEOMOD, new urban land is allocated to grids strictly following the order of suitability values from high to low, which is a deterministic approach²⁸. Seto et al.¹⁴ revised the original GEOMOD by allowing a certain degree of random perturbation in the allocation of new urban land.”

“The GGLM model²⁶, however, is for the simulation of global land use changes with multiple types (including urban land) under different future scenarios. The simulation is based on the downscaling of an existing future land product with a coarse resolution of 0.5 arc degree. The results of the GGLM model have a finer resolution of 1 km. The accuracies of the results depend largely on the original coarse-resolution future land product. In addition, in the GGLM model, the land type conversion of each grid is also determined by the land type with the highest conversion probability.”

“Our projections are generated using FLUS model coupled with the latest SSPs, which describe potential pathways in the coming century concerning policy assumptions and the socio-economic narrative, being consistent with the recent IPCC framework. These projections thus provide better potential in supporting the research in related disciplines, such as ecological protection, water security, urban climate and global climate change.”

We mentioned the issues of cross-scale dependencies as one of the limitations in our study:

“Third, urban development policies are usually operated at a country level, but in this study, we provide urban land demands at a regional scale, mainly due to the lack of country-level scenario data. We will try to solve this limitation in our future work by complementing more data to obtain the relevant country-level information for different scenarios.”

The idea to investigate cropland losses from urbanization is good, but to draw conclusions on crop production losses and affected people is quite misleading. First, there is no interaction between urbanization and cropland development modelled, e.g. cropland conversion to urban area might trigger expansion of cropland elsewhere. Second, the authors did not consider yield changes due to changes in agricultural management practices which might compensate the projected losses. Third, the number of affected people will strongly vary between world regions due to access to food and different diets.

Response:

In the revised paper, we supplement the analysis of the interaction between urban land and cropland, considering the land productivities in different scenarios:

“Despite the potential decline of urban population in some regions of the world, global urban land is expected to experience rapid growth before the 2040s. Our projections have demonstrated that most future urban land expansion occurs over croplands (50-63%) and forests (30-44%), causing the losses of food production and supply. To offset these losses, however, reclaiming cropland remains as a most immediate way, because land productivity may be difficult to improve substantially in the foreseeing future. The losses in croplands and forests will also have profound impacts on ecosystem services, such as carbon sequestration, habitat provision and food supply to human societies⁵⁵⁻⁵⁸.”

“Note that in this study we only account the direct impacts of urban expansion on other land types. To gain a clearer understanding, we complement our results with those obtained from Popp et al.²⁴, which provide scenario projections of cropland expansion on other land use types. Their results suggest that the SSP3 scenario features the largest amount of cropland expansion (approximately 700 million ha from 2005 to 2100), while the SSP1 scenario has the smallest increase of cropland area (approximately 20 million ha from 2005 to 2100). For SSP2, SSP4 and SSP5, the cropland areas are expected to increase between 250 million ha and 400 million ha from 2005 to 2100. Our scenario projections, which mainly focus on the direct impacts of urban expansion on cropland, reveal that SSP5 has the largest cropland loss due to urban growth (37.6 million ha), while SSP3 has the smallest loss (22.0 million ha). The estimated cropland losses in other SSP scenarios vary slightly from 22.4 million ha to 24.9 million ha. Therefore, the

combination of our and Popp's results indicates contrasting patterns of urban land and cropland expansion. In scenarios with more cropland losses caused by urban expansion, increases of cropland are relatively small, and vice versa."

In addition, we have described the method for estimating the affected population, the meaning of the estimated results, and the limitation of regardless of agricultural management practices in further detail:

"The diet structures of people vary largely from one place to another. To estimate the number of people affected by the global crop production losses, we use the global average per capita food supply of rice, wheat, maize, potatoes and vegetables as the basis for the estimation. These statistics can represent the overall situation of food supply worldwide and hence are adequate to apply in our analysis (<http://www.fao.org/faostat/en/>).

"The estimated number of people affected by the losses of crop production vary widely from 122 million to 1389 million, depending on which food consumption statistics are used (Table 2). Note that these results are not the numbers of people suffering from starving, but the maximum numbers of people affected by the production losses of a single food crop type, if these losses cannot be compensated with productions of other food crops. In addition, we did not consider yield changes due to changes in agricultural management practices which might compensate the projected losses."

The discussion is still imbalanced and should address the modelled urbanization trends and patterns in greater regional detail. Again, I would expect a more critical reflection of the consequences of these land-use changes not only on cropland losses but also on societies.

Response:

In the discussion section, we have provided the discussion of urban expansion and its consequences on a more detailed regional scale, comparing typical metropolitan areas and their regions.

"In addition to the regional-scale urban expansion projections as analyzed in the previous section, we also analyzed the urban expansion trends of the three typical international metropolitan areas under different SSPs, including New York, London and the Yangtze River Delta (Supplementary Figure 20). Although these three metropolitan areas are in countries with different development stages, their urban expansion trends are similar: By 2100, the urban area will become the largest in SSP5, followed by SSP1, SSP2, SSP4, and SSP3. The difference is that the urban expansion trends of New York and London in SSP2 are closer to those in SSP1, but the trend of the Yangtze River Delta in SSP2 is closer to that in SSP4. Moreover, the urban expansion in these selected metropolitan areas is evidently faster than that of their countries' average, except those in the developed countries in the SSP5 scenario (Supplementary Table 3). Therefore, rapid urban expansion may further increase the pressure on the environments and resources of these metropolitan areas which are of dense population."

Supplementary Figure 20 | Urban area growth in the three major metropolitan areas from 2015 to 2100. (a) London metropolitan; (b) New York metropolitan; (c) Yangtze River Delta metropolitan.

Supplementary Table 3 | The growth rate (%) of urban land area in the typical metropolitan areas and its regions by 2100

Metropolitan area / Region	SSP1	SSP2	SSP3	SSP4	SSP5
London	70.31	65.74	6.07	34.74	150.97
New York	95.68	90.35	19.83	55.49	171.19
Yanatzze River	55.98	46.81	34.68	46.79	61.39
EU-15	65.60	58.72	3.46	28.78	170.04
USA	76.61	70.33	12.36	39.14	189.81
China	32.55	27.29	20.54	26.64	36.84

Note: EU-15 represents the European Union member states that joined prior to 2004, including

Austria, Belgium, Denmark, Finland, France, Germany, Greece, Ireland, Italy, Luxembourg, Netherlands, Portugal, Spain, Sweden, and the United Kingdom.

** See Nature Research's author and referees' website at www.nature.com/authors for information about policies, services and author benefits

This email has been sent through the Springer Nature Tracking System NY-610A-NPG&MTS

Confidentiality Statement:

This e-mail is confidential and subject to copyright. Any unauthorised use or disclosure of its contents is prohibited. If you have received this email in error please notify our Manuscript Tracking System Helpdesk team at <http://platformsupport.nature.com> .

Details of the confidentiality and pre-publicity policy may be found here <http://www.nature.com/authors/policies/confidentiality.html>

Privacy Policy | Update Profile

DISCLAIMER: This e-mail is confidential and should not be used by anyone who is not the original intended recipient. If you have received this e-mail in error please inform the sender and delete it from your mailbox or any other storage mechanism. Springer Nature Limited does not accept liability for any statements made which are clearly the sender's own and not expressly made on behalf of Springer Nature Ltd or one of their agents.

Please note that Springer Nature Limited and their agents and affiliates do not accept any responsibility for viruses or malware that may be contained in this e-mail or its attachments and it is your responsibility to scan the e-mail and attachments (if any).

REVIEWERS' COMMENTS:

Reviewer #1 (Remarks to the Author):

Rereview of "Global Projections of Future Urban Land Expansion under Shared Socioeconomic Pathways"

The authors have made a good-faith effort to respond to all of my comments on the last draft of the manuscript. In particular, I am pleased they:

- 1.) Put their data in an online data repository with a DOI.
- 2.) Cited the Zhou et al. manuscript appropriately.
- 3.) Added a supplementary figure comparing their scenarios with other urban growth scenarios.

A few minor suggestions:

In the caption for Supplementary Figure 19, I suggest that the authors state that this is the absolute difference between the two models (I think that is correct, but am not certain).

Line 53, I believe it is correct to say "fossil-fuelled" rather than "fossil-fueled".

Reviewer #2 (Remarks to the Author):

In my opinion the authors did a very good job in improving their manuscript. The critical issues raised by the reviewers are addressed adequately. In particular the integration of a relatively simple but nevertheless sound uncertainty analysis and the more detailed comparison of their results are important new elements of their revised study. Therefore I recommend to publish the article as it is.

REVIEWERS' COMMENTS:

Reviewer #1 (Remarks to the Author):

Rereview of "Global Projections of Future Urban Land Expansion under Shared Socioeconomic Pathways"

The authors have made a good-faith effort to respond to all of my comments on the last draft of the manuscript. In particular, I am pleased they:

- 1.) Put their data in an online data repository with a DOI.
- 2.) Cited the Zhou et al. manuscript appropriately.
- 3.) Added a supplementary figure comparing their scenarios with other urban growth scenarios.

A few minor suggestions:

In the caption for Supplementary Figure 19, I suggest that the authors state that this is the absolute difference between the two models (I think that is correct, but am not certain).

Response:

We have changed the title of the original Supplementary Figure 19 to "The absolute difference between our product and other global urban land products in 2030".

Line 53, I believe it is correct to say "fossil-fuelled" rather than "fossil-fueled".

Response:

We have changed the word to "fossil-fuelled".

Reviewer #2 (Remarks to the Author):

In my opinion the authors did a very good job in improving their manuscript. The critical issues raised by the reviewers are addressed adequately. In particular the integration of a relatively simple but nevertheless sound uncertainty analysis and the more detailed comparison of their results are important new elements of their revised study. Therefore I recommend to publish the article as it is.